# Simultaneous cyclin D1 overexpression and p27[kip1] knockdown enable robust Müller glia cell cycle reactivation in uninjured mouse retina

Zhifei Wu[1†], Baoshan Liao[1†], Julia Ying[1], Jan Keung[1,2], Zongli Zheng[1,2], Virpi Ahola[2,3], Wenjun Xiong[1]*

[1]Department of Biomedical Sciences and Tung Biomedical Sciences Centre, City University of Hong Kong, Hong Kong, China; [2]Ming Wai Lau Centre for Reparative Medicine, Karolinska Institutet, Hong Kong, China; [3]Institute of Biomedicine, University of Eastern Finland, Kuopio, Finland

*For correspondence:
wenjun.xiong@cityu.edu.hk

[†]These authors contributed equally to this work

## eLife Assessment

This manuscript presents a potentially **important** strategy for stimulating mammalian Müller glia to proliferate in vivo by manipulating cell cycle components. The results are **convincing** that a large number of Müller glia can be induced to re-enter the cell cycle without a damage stimulus. These findings are likely to appeal to retinal biologists and neuroscientists in general.

**Abstract** Harnessing the regenerative potential of endogenous stem cells to restore lost neurons is a promising strategy for treating neurodegenerative disorders. Müller glia (MG), the primary glial cell type in the retina, exhibit extraordinary regenerative abilities in zebrafish, proliferating and differentiating into neurons post-injury. However, the regenerative potential of mouse MG is limited by their inherent inability to re-enter the cell cycle, constrained by high levels of the cell cycle inhibitor p27[Kip1] and low levels of cyclin D1. Here, we report a method to drive robust MG proliferation by adeno-associated virus (AAV)-mediated cyclin D1 overexpression and p27[Kip1] knockdown. MG proliferation induced by this dual targeting vector was self-limiting, as MG re-entered cell cycle only once. As shown by single-cell RNA-sequencing, cell cycle reactivation led to suppression of interferon signaling, activation of reactive gliosis, and downregulation of glial genes in MG. Over time, the majority of the MG daughter cells retained the glial fate, resulting in an expanded MG pool. Interestingly, about 1% MG daughter cells expressed markers for retinal interneurons, suggesting latent neurogenic potential in a small MG subset. By establishing a safe, controlled method to promote MG proliferation in vivo while preserving retinal integrity, this work provides a valuable tool for combinatorial therapies integrating neurogenic stimuli to promote neuron regeneration.

## Introduction

Müller glia (MG) are the last cell type generated by the retinal progenitor cells (RPCs) during development and exhibit a gene expression profile similar to that of late RPCs (*Jadhav et al., 2009*; *Roesch et al., 2012*). In teleost fish and amphibians, MG respond rapidly to retinal injury, undergoing robust proliferation and regenerating lost retinal neurons from MG-derived progenitor cells (*Hamon et al., 2016*; *Todd and Reh, 2022b*). In contrast, the proliferative and neurogenic ability of MG in response to injury in mammals is severely limited, failing to mediate retinal self-repair (*Karl et al., 2008*). Recent

investigations have achieved great success in mammalian retinal regeneration by stimulating MG reprogramming through a single or combination of transcription factors (*Jorstad et al., 2017*; *Le et al., 2024a*; *Le et al., 2024b*; *Todd et al., 2021*; *Todd et al., 2022a*). Other studies have shown that the quiescent state of MG can be overridden by upstream signaling pathways such as Wnt and Hippo (*Yao et al., 2016*; *Hamon et al., 2019*; *Rueda et al., 2019*). Activation of Wnt signaling by forced expression of β-catenin in adult mouse MG promoted spontaneous cell cycle re-entry in uninjured retinas (*Yao et al., 2016*). Moreover, it was demonstrated that bypassing the Hippo pathway in mouse MG led to spontaneous re-entry into the cell cycle and reprogramming into a progenitor cell-like state (*Hamon et al., 2019*; *Rueda et al., 2019*). These findings suggest that the re-entry of MG into the cell cycle, which is the first step of MG-mediated retinal regeneration in zebrafish, could be unlocked in mammalian retinas.

The cell cycle of MG is mainly regulated by cyclins and cyclin-dependent kinases (CDKs). Cyclins bind to CDKs to form cyclin-CDK complexes to promote cell cycle progression. During retinal development, the expression of D-type cyclins (cyclins D1, D2, and D3) is tightly regulated (*Barton and Levine, 2008*; *Dyer and Cepko, 2001*; *Trimarchi et al., 2008*). Among these, cyclin D1, encoded by the *Ccnd1* gene, is the predominant D-type cyclin in the developing retina and is highly expressed in the RPCs but absent in differentiated cells (*Barton and Levine, 2008*; *Trimarchi et al., 2008*). Mice lacking *Ccnd1* have small eyes and hypocellular retinas due to reduced RPC proliferation (*Fantl et al., 1995*; *Sicinski et al., 1995*), which cannot be compensated by *Ccnd2* and *Ccnd3* (*Das et al., 2012*; *Das et al., 2009*). Negative regulators of the cell cycle are CDK inhibitors (CDKIs), which include the INK4 (p16$^{INK4a}$, p15$^{INK4b}$, p18$^{INK4c}$, and p19$^{INK4d}$) and CIP/KIP families (p21$^{Cip1}$, p27$^{Kip1}$, and p57$^{Kip2}$) (*Reynisdóttir et al., 1995*). The CDKIs inhibit cell cycle progression by binding to and inactivating the cyclin-CDK complexes (*Besson et al., 2008*), and they regulate the proliferation in distinct RPCs (*Dyer and Cepko, 2000*; *Dyer and Cepko, 2001*; *Levine et al., 2000*). P27$^{Kip1}$ inhibits the cyclin D-CDK complex to enter the S phase. Following acute retinal damage in mice, a very small number of MG re-enter the cell cycle, coincident with the downregulation of p27$^{Kip1}$ or upregulation of cyclin D1 (*Dyer and Cepko, 2001*; *Hamon et al., 2019*; *Rueda et al., 2019*; *Yao et al., 2016*). However, this process is transient, as cyclin D1 expression rapidly returns to the basal level (*Hamon et al., 2019*; *Rueda et al., 2019*).

In this study, we propose that targeting the two key regulators of the cell cycle, cyclin D1 and p27$^{kip1}$, can effectively induce MG cell cycle re-entry. Our findings demonstrate that simultaneously reducing p27$^{Kip1}$ and increasing cyclin D1 in MG using a single AAV vector has a strong synergistic effect on promoting MG proliferation in uninjured adult mouse retinas. MG proliferation induced by this treatment is robust and self-limiting, as MG undergo a single round of cell division rather than unlimited proliferations. Through single-cell RNA sequencing (scRNA-seq), we observed that cell cycle reactivation leads to the downregulation of the interferon (IFN) pathways in the MG and suppression of the MG genes. By RNA in situ hybridization and immunostaining, we showed that MG partially and temporally suppressed their glial cell fate, while the majority of MG regained the normal glial identify by 4 months post-CCA treatment. A few EdU+ MG daughter cells in the inner nuclear layer (INL) express the bipolar cell marker Otx2 or the amacrine cell marker HuC/D, suggesting rare de novo neurogenesis from MG. Importantly, MG cell cycle reactivation does not disrupt retinal structure or impair retinal function, as the treatment did not deplete the MG from the retina or cause neoplasia. In summary, our results showed that downregulating p27$^{Kip1}$ and upregulating cyclin D1 stimulate MG proliferation, and it is possible to combine this approach with other factors that promote regeneration to enhance retinal repair mediated by MG.

## Results

### Simultaneous p27$^{Kip1}$ downregulation and cyclin D1 overexpression drive robust MG proliferation in the uninjured mouse retina

To test the hypothesis that adult mouse MG are kept in a quiescent state by high levels of p27$^{Kip1}$ and low levels of cyclin D1, we examined whether spontaneous MG proliferation could be activated by directly changing the levels of these two downstream cell cycle regulators. To drive MG-specific gene expression, adeno-associated virus (AAV)-mediated transgene expression was controlled by a promoter sequence cloned from the human *glial fibrillary acidic protein* (*GFAP*) gene (*Figure 1A*,

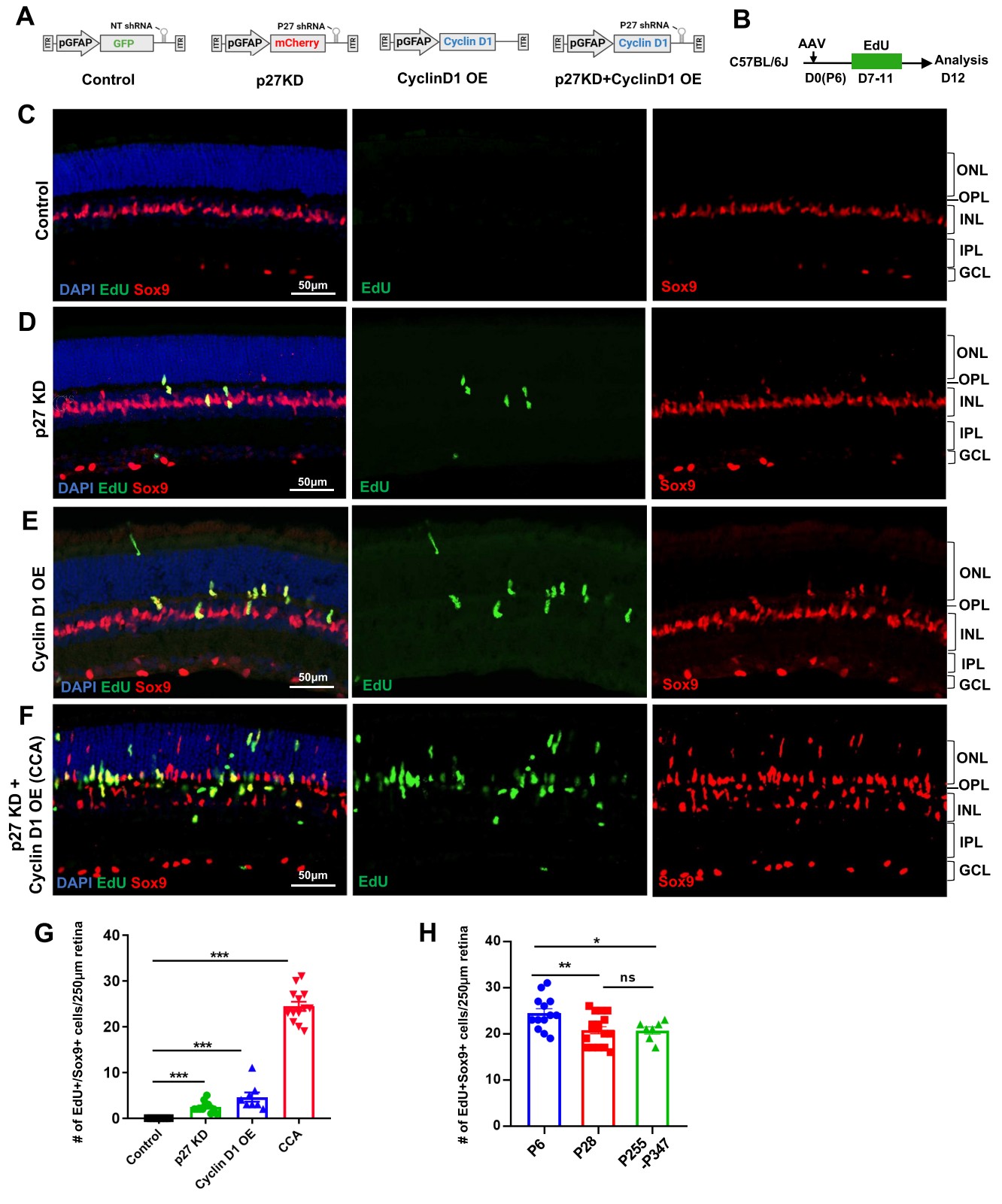

**Figure 1.** Simultaneous p27[Kip1] downregulation and cyclin D1 overexpression drive robust Müller glia (MG) proliferation in the uninjured mouse retina. (**A**) Schematic representations of adeno-associated virus (AAV) vectors used in this study. AAV-GFAP-GFP-non-target (NT) shRNA for control, AAV-GFAP-mCherry-p27 shRNA for p27[kip1] knockdown (KD), AAV-GFAP-cyclin D1 for cyclin D1 overexpression (OE), and AAV-GFAP-cyclin D1-p27 shRNA for p27[kip1] KD and cyclin D1 OE. (**B**) Experimental design. Mice received an intravitreal AAV injection on postnatal day 6 (P6), designated as D0, and daily 5-ethynyl-

*Figure 1 continued on next page*

*Figure 1 continued*

2'-deoxyurdine (EdU) injections intraperitoneally from D7 to D11. (**C–F**) Analysis of EdU incorporation with Sox9 co-labeling in uninjured mouse eyes injected with indicated viruses. ONL, outer nuclear layer; OPL, outer plexiform layer; INL, inner nuclear layer; IPL, inner plexiform layer; GCL, ganglion cell layer. (**G**) Quantification of EdU+Sox9+ cells per 250 μm high infection areas. Control (n=8 eyes), p27$^{kip1}$ KD (n=11 eyes), cyclin D1 OE (n=8 eyes), CCA (n=14 eyes). (**H**) Quantification of EdU+Sox9 + cells per 250 μm high infection areas in retinas injected with CCA at indicated ages: P6 (n=13), P28 (n=17), and P255-P347 (n=7). Five EdU injections were given intraperitoneally from D7 to D11 post-CCA injection, and retinas were harvested for EdU analysis on D12. Data are presented as mean ± SEM. *p<0.05, **p<0.01, ***p<0.001, ns = not significant (one-way ANOVA with Tukey's post hoc test) (**G–H**).

The online version of this article includes the following source data and figure supplement(s) for figure 1:

**Figure supplement 1.** Specific GFP expression in Müller glia (MG) of mouse eyes intravitreally injected with AAV-GFAP-GFP vector.

**Figure supplement 2.** Downregulation of p27$^{kip1}$ expression in Müller glia (MG) by AAV-GFAP-mCherry-p27 shRNA1/2 vector.

**Figure supplement 3.** Müller glia (MG)-specific upregulation of cyclin D1 in mouse retina infected with AAV-GFAP-cyclin D1.

**Figure supplement 3—source data 1.** PDF file containing original western blots for panel C, indicating the relevant bands and virus treatment.

**Figure supplement 3—source data 2.** Original files for western blots displayed in panel C.

**Figure supplement 4.** Infection pattern of intravitreally delivered adeno-associated virus (AAV) vector.

**Figure supplement 5.** Comparison of cyclin D1 overexpression and p27$^{kip1}$ knockdown efficiencies by different viruses.

**Figure supplement 6.** Characterization of Müller glia (MG) proliferation in *Glast$^{Cre-ERT2}$; Rosa26$^{LSL-Sun1:GFP}$* transgenic mice.

*Figure 1—figure supplement 1A*; *Lee et al., 2008*). AAV serotype 7m8 vectors were injected intravitreally into the eyes of C57BL/6J mice on postnatal day 6 (P6), when the majority of MG are born and start differentiation, and GFP reporter expression could be detected at 3 days post-AAV injection (*Figure 1—figure supplement 1B*). Our close examination showed that this *GFAP* promoter drives transgene expression specifically in MG but not in astrocytes (*Figure 1—figure supplement 1C–F*).

To examine the effect of p27$^{Kip1}$ and cyclin D1 levels on MG proliferation, wild-type mice were infected with control AAV or AAV that knocked down p27$^{Kip1}$ and/or overexpressed cyclin D1 at P6 and received intraperitoneal injections of 5-ethynyl-2'-deoxyurdine (EdU) for 5 consecutive days, from P13 to P17, to label the MG that had entered the S phase of the cell cycle (*Figure 1A and B*). In the control retinas that were infected by AAV$_{7m8}$-GFAP-GFP-non-target (NT) shRNA, all MG, which were positive of Sox9 expression, were negative of EdU labeling (*Figure 1C*). When the retina was infected by AAV$_{7m8}$-GFAP-mCherry-p27 shRNA1, which expressed a highly efficient *p27* shRNA1, a small number of MG cells re-entered the cell cycle (*Figure 1D and G*, *Figure 1—figure supplement 2*); however, the vast majority of MG remained in a quiescent state. Overexpressing cyclin D1 alone through AAV$_{7m8}$-GFAP-cyclin D1 infection stimulated a subset of MG cells to proliferate, resulting in a threefold increase in the number of EdU+ MG cells compared to p27$^{Kip1}$ knockdown (*Figure 1E and G*, *Figure 1—figure supplement 3*). Due to the uneven efficiency of AAV infection across the retina, quantification was performed in the region with the highest infection levels (*Figure 1—figure supplement 4*). Finally, the AAV$_{7m8}$-GFAP-cyclin D1-p27 shRNA1 vector, which simultaneously overexpressed cyclin D1 and suppressed p27$^{Kip1}$, had the most significant impact on MG proliferation, with a fivefold increase in EdU+Sox9+ cells compared to cyclin D1 overexpression alone (*Figure 1F and G*). The differences in MG proliferation across groups were not due to variations in virus infection efficacy, as confirmed by measuring p27$^{Kip1}$ knockdown and cyclin D1 overexpression levels by quantitative PCR (*Figure 1—figure supplement 5*). A transgenic mouse line *Glast$^{Cre-ERT2}$; Rosa26$^{LSL-Sun1:GFP}$*, in which MG nuclei were labeled by nuclear membrane-bound GFP (*Figure 1—figure supplement 6A–C*), was used to quantify the percentage of MG that re-entered the cell cycle. In the retinal area where viral infection rate was the highest, approximately 45% of Sun1:GFP+ MG were EdU positive, and the total number of MG increased by about 50% (*Figure 1—figure supplement 6D–H*), indicating that nearly half of the MG cells re-entered the cell cycle. As the AAV$_{7m8}$-GFAP-cyclin D1-p27 shRNA1 vector enables such robust MG proliferation, we refer to it as the cell cycle activator (CCA) for short.

Previous research has shown that the ability of retina to regenerate by various stimuli declines with the age of the mice (*Löffler et al., 2015*; *Ueki et al., 2015*). We compared the efficiency of CCA in driving MG proliferation in young (P28) and older adult mouse (P255-P347) retinas to that in P6 pups (*Figure 1H*). Remarkably, MG proliferation induced by CCA remained robust in adult mice (*Figure 1H*). These findings suggest that CCA efficiently drives MG proliferation irrespective of the age of the mice. The synergistic effect of cyclin D1 overexpression and p27$^{Kip1}$ knockdown on MG

proliferation suggest that both low levels of p27$^{Kip1}$ and high levels of cyclin D1 are required for post-mitotic MG to re-enter the cell cycle.

## MG proliferation driven by CCA is self-limiting

Concerned that p27$^{Kip1}$ suppression and cyclin D1 overexpression may lead to uncontrolled cell proliferation and retinal tumorigenesis, we analyzed the proliferative capacity of MG driven by CCA. First, we examined the duration of MG proliferation after CCA treatment by a time-course EdU incorporation assay. Starting on various days after the CCA treatment, mice received two injections of EdU to label the cells that were undergoing proliferation (*Figure 2A*). The result revealed that MG proliferation started as quickly as the third day after CCA injection, reached its peak around the fifth day, followed by a gradual decrease (*Figure 2A*). By 2 weeks post-CCA injection, only a few MG were observed re-entering the cell cycle. Two months post-CCA injection, MG proliferation had mostly ceased (*Figure 2A*). These findings suggest that MG proliferation was largely completed within 2 weeks post-CCA treatment. In addition, an EdU/5-bromo-2' deoxyuridine (BrdU) double-labeling assay was performed to examine whether MG undergoes one or multiple cell divisions after CCA treatment. Five injections of EdU were given from day 1 to 5 post-CCA treatment, followed by five BrdU injections from day 6 to 10 post-CCA treatment (*Figure 2B*). Retinas were collected 1 day after the last BrdU injection to evaluate if any MG continuously entered the S phase of the cell cycle. While there were a number of cells positive for EdU or BrdU, no cells were co-labeled with EdU and BrdU (*Figure 2B and C*), indicating that no MG underwent two cell divisions. Finally, we utilized the *Glast$^{Cre-ERT2}$*; *Rosa26$^{LSL-tdTomato}$* mouse line to label MG sparsely with a low dose of tamoxifen induction (*Figure 2D*, *Figure 2—figure supplement 1*). At 4 weeks post-CCA injection, we observed either single MG or pairs of MG in close proximity, but no clusters of three or more cells (*Figure 2E and F*). The results of these experiments suggest that MG undergo only one cell division following CCA treatment.

We asked why CCA did not drive MG to undergo multiple rounds of cell proliferation. We immunostained the retinal samples harvested for the time-course MG proliferation study (*Figure 2A*). Cyclin D1 was undetectable in MG of control retinas (*Figure 2G*), but it was robustly overexpressed in the majority of MG within the INL by 3 days post-CCA injection (*Figure 2H*). By a later time point, cyclin D1 expression had disappeared in the EdU+ MG that had migrated to the ONL (*Figure 2I*, arrowheads). The other EdU– MG in the ONL, which were likely the cells that have proliferated prior to EdU injections, also ceased to express cyclin D1 (*Figure 2I*). In another group of mice injected with CCA on P28, we further confirmed that the levels of cyclin D1 overexpression decreased at 3 weeks post-CCA injection and diminished at 4 months post-CCA injection (*Figure 2—figure supplement 2*). In comparison, p27$^{Kip1}$ suppression by CCA lasted longer, as p27$^{Kip1}$ level remained low in some MG at 3 weeks and 4 months post-CCA injection (*Figure 2—figure supplements 3 and 4*). These results suggest that cyclin D1 overexpression ceased after the initial cell proliferation, thereby preventing MG from undergoing a second round of proliferation.

## scRNA-seq analysis of MG shows suppression of the IFN pathway by CCA

To identify changes in gene expression and possible changes in cell fate of MG and their daughter cells, we performed scRNA-seq analysis on the mouse retinas that received CCA treatment. To isolate MG in regions of the retina with high viral infection, CCA and a control virus AAV$_{7m8}$-GFAP-GFP-NT shRNA were 9:1 mixed and co-injected to 4-week-old *Glast$^{Cre-ERT2}$*; *Rosa26$^{LSL-tdTomato}$* mice (*Figure 3A*). Three weeks post-infection, the retina areas with strong GFP expression were collected by dissection, and tdTomato+ MG were isolated by fluorescence-activated cell sorting (FACS) and analyzed by scRNA-seq (*Figure 3A*). The majority of sorted MG should be infected by CCA. The control group was injected with the AAV$_{7m8}$-GFAP-GFP-NT shRNA virus only (the same total concentration as the CCA group) to account for any nonspecific effect caused by virus injection and/or shRNA expression. As previous studies demonstrated that *N*-methyl-D-aspartate (NMDA)-induced retinal damage and histone deacetylase inhibitor trichostatin A (TSA) improve MG proliferation and reprogramming (*Hoang et al., 2020*; *Jorstad et al., 2017*; *Rueda et al., 2019*), a group of CCA-treated mice also received NMDA on day 7 and TSA on day 9 post-CCA injection (CCA+NMDA+TSA, referred to as CCANT) to enhance the reprogramming effect, if any, of CCA.

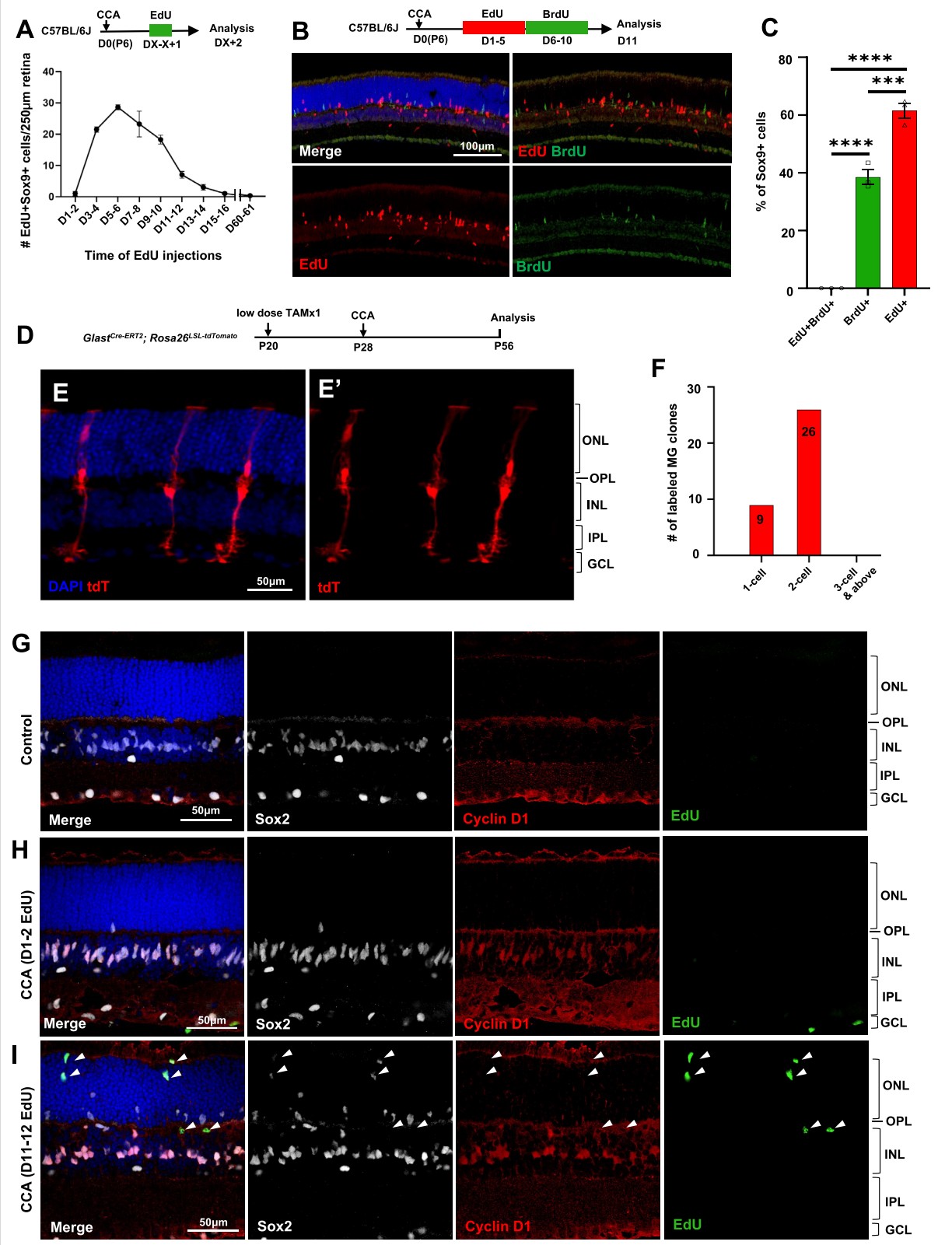

**Figure 2.** Müller glia (MG) proliferation induced by cell cycle activator (CCA) is self-limiting. (**A**) Time-course analysis of MG proliferation following CCA injection. 5-Ethynyl-2′-deoxyurdine (EdU) was administered for 2 consecutive days, starting at various days post-CCA injection, with samples harvested 1 day after the second EdU injection. Data are presented as mean ± SEM (n≥4). (**B**) Analysis of cells labeled with EdU and 5-bromo-2' deoxyuridine (BrdU). (**C**) Quantification of the percentages of EdU+BrdU−, EdU−BrdU+, and EdU+BrdU+ cells of the total Sox9+ cells. Data are presented as mean ± SEM

*Figure 2 continued on next page*

*Figure 2 continued*

(n=3). ***p<0.001, ****p<0.0001 (one-way ANOVA with Tukey's post hoc test). (**D**) Experimental design. (**E**) A representative image of sparsely labeled MG in the *Glast*<sup>Cre-ERT2</sup>; *Rosa26*<sup>LSL-tdTomato</sup> mouse retina. (**F**) Quantification of the numbers of 1-cell, 2-cell, and larger clones in ten 250 μm hotspot areas from four retinas infected with CCA. (**G–I**) Eye samples from the time-course EdU analysis (**A**) were stained for cyclin D1 and Sox2. Representative retinal sections from uninjected eyes (**G**), mice with EdU injections at D1–2 and harvest at D3 (**H**), and mice with EdU injections at D11–12 and harvest at D13 (**I**). Arrowheads point out the EdU+ cells that are negative for cyclin D1 staining. ONL, outer nuclear layer; OPL, outer plexiform layer; INL, inner nuclear layer; IPL, inner plexiform layer; GCL, ganglion cell layer.

The online version of this article includes the following figure supplement(s) for figure 2:

**Figure supplement 1.** Characterization of Müller glia (MG) labeling and leaky expression in *Glast*<sup>Cre-ERT2</sup>; *Rosa26*<sup>LSL-tdTomato</sup> transgenic mice.

**Figure supplement 2.** Analysis of cyclin D1 expression at different days post-cell cycle activator (CCA) injection on P28.

**Figure supplement 3.** Analysis of p27<sup>kip1</sup> expression at different days post-cell cycle activator (CCA) injection on P6.

**Figure supplement 4.** Analysis of p27<sup>kip1</sup> expression at different days post-cell cycle activator (CCA) injection on P28.

After quality control, 3758 cells were profiled in the control group, 3890 cells in the CCA group, and 3278 cells in the CCANT group (*Figure 3—figure supplements 1 and 2*). Clustering analysis separated the cells into six distinct clusters (*Figure 3B*), which are quiescent MG, reactivated MG, MG in G2/M phase, and MG in S phase, rods, and rod-MG, as annotated by the known retinal cell-type markers (*Figure 3C*). The rods and rod-MG clusters were due to tdTomato leaky expression in native rods and rod mRNA contamination during MG isolation, respectively (*Figure 3—figure supplements 3 and 4*). The vast majority of MG (>90%) in the control group are quiescent MG, which express high levels of MG genes such as *Glul* and *Kcnj10* (*Figure 3D–G*). In the CCA-treated sample, there was still a small number of proliferating MG in the G2/M or S phase at 3 weeks after CCA treatment, as shown by cycle state analysis (*Figure 3B–E*, *Figure 3—figure supplement 2D and E*). ~70% of MG in the CCA-treated sample formed a separate cluster, which we refer to as reactivated MG (*Figure 3C–E*). Reactive gliosis genes (*Gfap* and *Vim*) were upregulated while other MG genes (*Kcnj10*, *Glul*, *Rlbp1*, and *Aqp4*) were downregulated in the reactivated MG compared with the quiescent MG (*Figure 3—figure supplement 5*). Interestingly, the top differentially expressed genes between reactivated MG and quiescent MG are the IFN pathway genes, including *Stat1*, *Stat2*, *Gbp6*, *Irgm1,* and *Igtp*, which were downregulated in the reactivated MG (*Figure 3F–H*). IFN signaling is involved in antiviral responses and usually induces cell cycle arrest in infected cells (*Durbin et al., 1996*; *Kaplan et al., 1998*). The high level of IFN signaling in the control sample likely resulted from control virus infection, while the pro-proliferative effect of the CCA vector might suppress IFN pathway to promote MG proliferation. In contrast, *Stat3*, which is often activated by various cytokines or growth factors to promote cell proliferation and survival (*Hirano et al., 2000*), was not downregulated concurrently with *Stat1* and *Stat2* (*Figure 3H*). Activation of STAT3 signaling in the reactivated MG may also facilitate MG proliferation.

## CCA causes a temporary suppression of MG genes, leading to partial dedifferentiation

Despite CCA or CCANT treatment, no neurogenic progenitor cluster or significant upregulation of neurogenic transcription factors such as *Ascl1* and *Neurog2* were observed (*Figure 3—figure supplement 6*), suggesting cell cycle reactivation, even with NMDA and TSA, is insufficient to drive robust neuronal reprogramming. Nonetheless, the scRNA-seq data suggest that CCA treatment led to downregulation of the MG genes, including *Glul*, *Rlbp1*, *Aqp4*, and *Kcnj10* (*Figure 3—figure supplement 5*, *Supplementary file 1*). To further verify the change of MG gene expression after CCA treatment, we assessed the level of *Glul,* the gene encoding glutamate synthase, which is highly expressed in the MG, by RNA in situ hybridization on retinal sections (*Figure 4A*). Three weeks post-CCA treatment, there was a significant decrease of *Glul* mRNA level in the MG, regardless their localization in the ONL, OPL, or INL (*Figure 4B–D*), suggesting a partial repression of the MG gene. By 4 months post-CCA treatment, the *Glul* level increased again, suggesting that they eventually retained MG identity and functions (*Figure 4B–D*).

We further assessed another MG marker Sox9 in the CCA-treated retinas. Immunostaining confirmed that many MG in the ONL, which have gone interkinetic nuclear migration, decreased or even lost Sox9 expression level at 3 weeks post-CCA treatment (*Figure 5A–E*). The suppression of MG

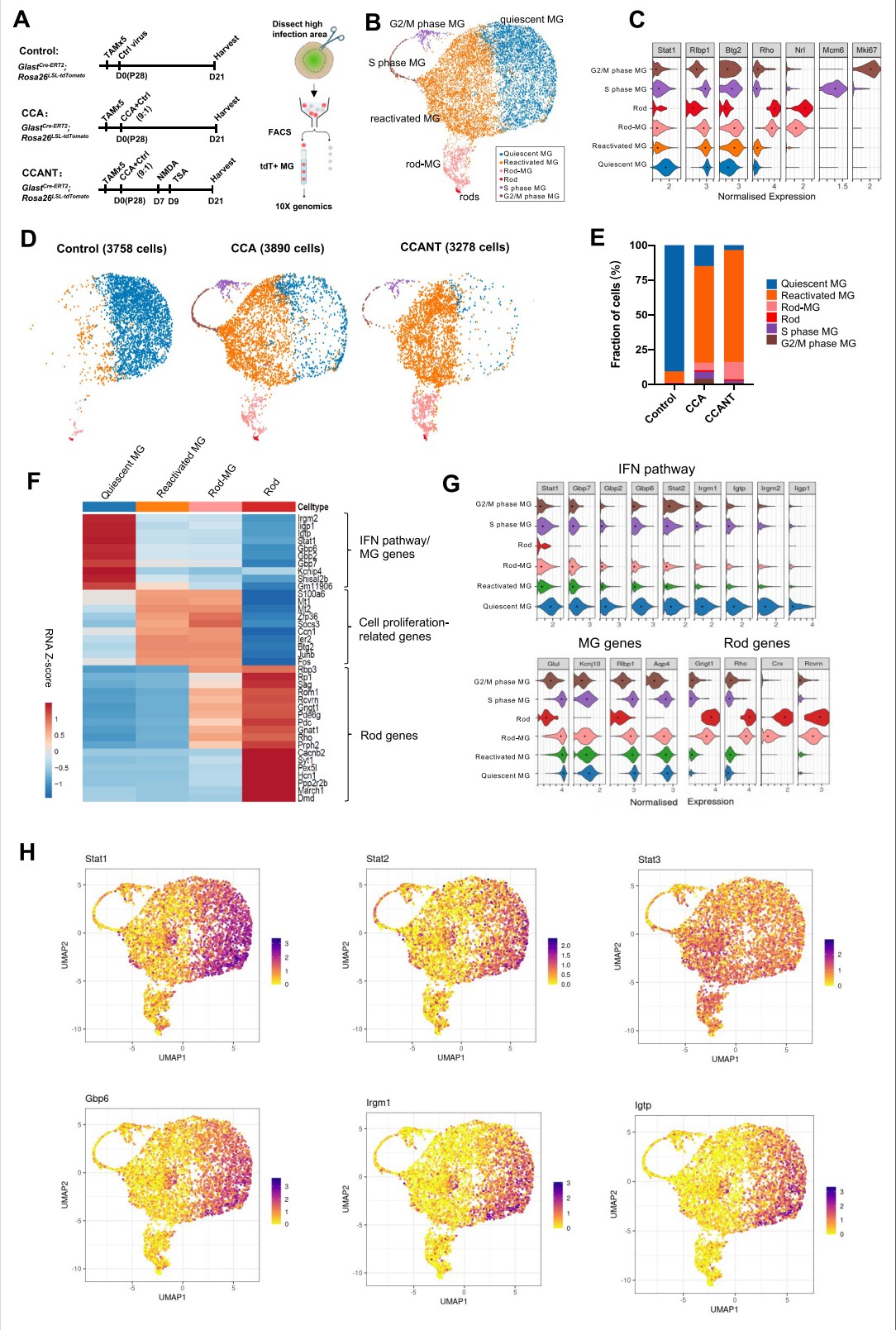

**Figure 3.** Single-cell RNA (scRNA-seq) analysis of Müller glia (MG) at 3 weeks post-CCA treatment. (**A**) Schematic illustration of the scRNA-seq experiment. (**B**) Uniform Manifold Approximation and Projection (UMAP) plot of scRNA-seq data for all MG combined from three groups with control, CCA, or CCANT treatment, with clusters identified based on known marker gene expression. (**C**) Violin diagram showing expression of retinal cell markers in different cell clusters. (**D**) Split UMAP plots of control, CCA, and CCANT groups. (**E**) Proportions of cell clusters within control, CCA, and

*Figure 3 continued on next page*

*Figure 3 continued*

CCANT groups. (**F**) Heatmap of top differentially expressed genes (DEGs) between cell clusters. Cell clusters are shown in columns, and genes are in rows. Color scale denotes Z score of the normalized gene expression levels. (**G**) Violin diagram illustrating the expression of interferon (IFN) pathway genes, MG genes, and rod genes across different cell clusters. (**H**) Feature plots showing normalized gene expression of *Stat1, Stat2, Stat3, Gbp6, Irgm1*, and *Igtp* in different cell clusters. CCA, cell cycle activator; CCANT, CCA+NMDA+TSA.

The online version of this article includes the following figure supplement(s) for figure 3:

**Figure supplement 1.** Preprocessing and filtering of single-cell RNA (scRNA) data and removal of doublet cells.

**Figure supplement 2.** Removal of possibly contaminated cells.

**Figure supplement 3.** Characterization of the rod-Müller glia (MG) and rod clusters in the single-cell RNA (scRNA-seq) analysis.

**Figure supplement 4.** RNA in situ hybridization analysis of rod gene expression in dissociated Müller glia (MG) following cell cycle activator (CCA) treatment.

**Figure supplement 5.** Expression level changes of Müller glia (MG) genes in reactivated MG.

**Figure supplement 6.** Absence of neurogenic progenitor clusters induced by cell cycle activator (CCA) treatment.

gene expression was transient, as most MG had recovered high levels of Sox9 expression by 4 months (*Figure 5E*). The results from both assays suggest that MG cell fate was partially suppressed in the MG daughter cells, but it recovered without further stimulus to drive neurogenesis. However, less than 1% of the GFP+ MG in the INL completely lost Sox9 between 3 weeks and 4 months post-CCA treatment (*Figure 5D and E*). These Sox9-negative MG exhibited circular nuclear envelop shape and faint GFP signal (*Figure 5D*, arrowhead), characteristics of the MG-derived retinal neuron-like cells observed in previous studies (*Hoang et al., 2020*; *Le et al., 2024a*; *Le et al., 2024b*).

## Rare neurogenesis from MG occurs spontaneously after cell cycle reactivation

To examine whether some MG, although at a very low efficiency, give rise to neurons after CCA treatment, we performed immunostaining for the bipolar marker Otx2. We focused our analysis on EdU+tdT+ MG in the *Glast^Cre-ERT2*; *Rosa26^LSL-tdTomato* mice to identify de novo neurogenesis from MG (*Figure 6A*). Four months post-CCA treatment, we found a small number of MG-derived cells in the INL expressing Otx2 (*Figure 6B–E*). Although their proportion was low (~1% of EdU+tdT+ cells), these were likely the genuine neuron-like cells that have differentiated from MG daughter cells over time, as these cells were not present at an earlier time point (*Figure 6A–E*). Rare HuC/D+EdU+tdT+ cells were also found in the lower INL, where the amacrine cells naturally reside, only at 4 months post-CCA treatment (*Figure 6—figure supplement 1*). However, the HuC/D level in the MG-derived cells was lower compared to that of the native amacrine cells (*Figure 6—figure supplement 1*). Whether these cells express other bipolar or amacrine cell markers and whether they are functional interneurons that connect with the retinal circuitry needs further investigation. Nonetheless, our findings suggest that cell cycle reactivation alone allows some MG daughter cells to spontaneously become neuron-like cells.

## CCA does not cause retinal neoplasia or functional deficit

To assess the long-term effect of CCA treatment, a cohort of C57BL/6J mice were observed for a year following CCA injection (*Figure 7A*). Visual function assessments showed no significant differences in visual acuity and electroretinography (ERG) between CCA-treated eyes and uninjected control eyes 1 year after CCA treatment (*Figure 7B–D*). Upon examining all the harvested retinas from the CCA-treated group, no retinal neoplasia was observed. The retinal structure remained intact, with no malignancy or disruptions in the retinal layers or stratification (*Figure 7E–H*). These results indicate that CCA treatment did not have a detrimental impact on retinal structure or function in mice, nor did it induce retinal tumors.

We used the MG marker Sox9 to examine the MG population in the retina sections harvested at a year post-CCA treatment. In the control retina, Sox9+ MG cells were aligned in the INL (*Figure 7E and F*). In the CCA-treated retinas, there was a significant expansion in the number of Sox9+ MG cells, distributed across the ONL, OPL, and INL (*Figure 7G and H*). The larger population of Sox9+ MG remained to support the retinal structure and homeostasis, which explains the relatively unaffected retinal structure and function observed on the CCA-treated eyes. We further made comparison of the

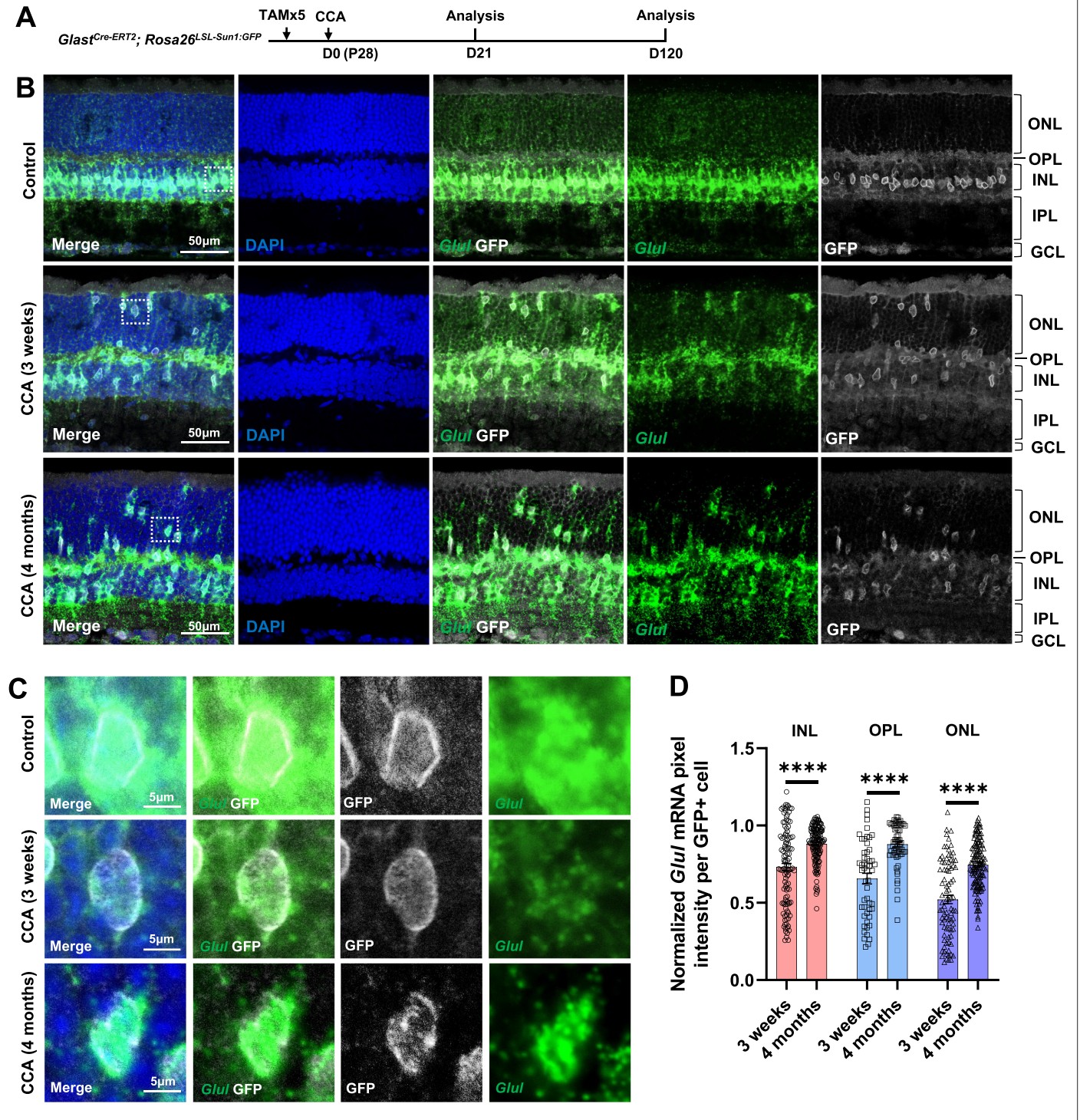

**Figure 4.** *Glul* mRNA levels decrease in the Müller glia (MG) that migrated to the ONL and OPL. (**A**) Experimental design. (**B**) In situ hybridization showing *Glul* mRNA expression. Control retina did not receive any adeno-associated virus (AAV) injection. ONL, outer nuclear layer; OPL, outer plexiform layer; INL, inner nuclear layer; IPL, inner plexiform layer; GCL, ganglion cell layer.(**C**) Magnified views of the highlighted cells in panel (**B**). (**D**) Average pixel intensity of *Glul* mRNA per GFP+ cell. Pixel intensity of MG treated with cell cycle activator (CCA) was normalized to the average pixel intensity of MG in the uninjected eye of the same animal. n=60 GFP+ cells in each retinal layer from three mice, data are presented as mean ± SEM. ****p<0.0001 (unpaired two-tailed Student's t-test) (**D**).

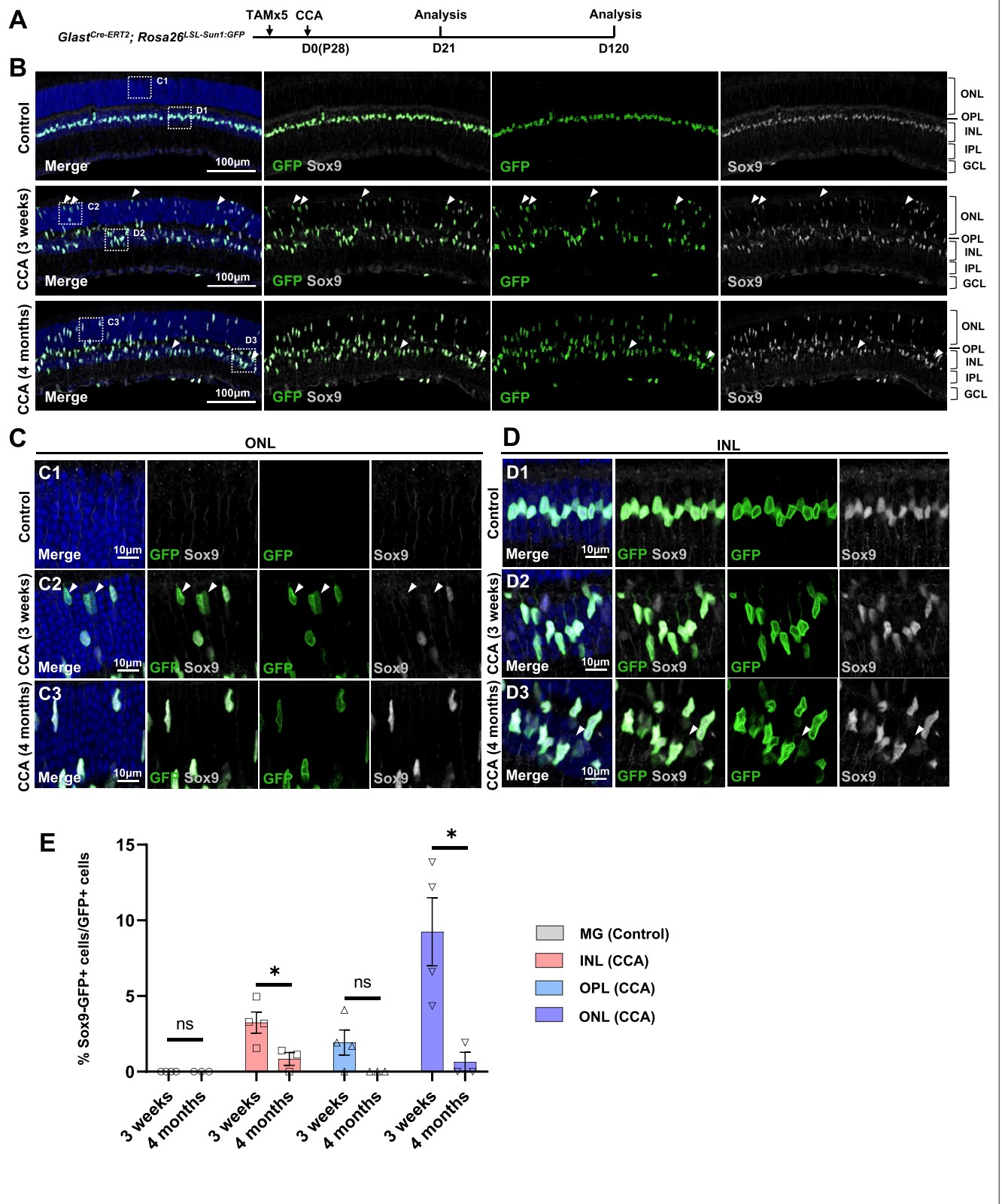

**Figure 5.** Temporary loss of Sox9 in a subset of Müller glia (MG) following cell cycle activator (CCA) treatment. (**A**) Experimental design. (**B**) Representative retinal sections of *Glast^Cre-ERT2^; Rosa26^LSL-Sun1:GFP^* mice without virus injection (control) or mice at 3 weeks and 4 months post-CCA injection. Arrowheads highlight Sox9−GFP+ cells. (**C–D**) Magnified views of the boxed regions in (**B**). Arrowheads highlight Sox9−GFP+ cells. (**E**) Quantification of Sox9−GFP+ cells as a percentage of total GFP+ cells in each retinal layer. n=3 mice, data are presented as mean ± SEM. ns =

*Figure 5 continued on next page*

*Figure 5 continued*

not significant, *p<0.05 (unpaired two-tailed Student's t-test). ONL, outer nuclear layer; OPL, outer plexiform layer; INL, inner nuclear layer; IPL, inner plexiform layer; GCL, ganglion cell layer.

numbers of Sox9+ cells after 1 year of CCA treatment and after 2 weeks of CCA treatment. The total numbers of Sox+ cells were similar between these two time points (*Figure 7I*), indicating no significant MG loss after MG proliferation. To directly assess whether CCA treatment may lead to the cell death of the MG, especially the MG displaced in the ONL and OPL, we examined the CCA-treated retinas

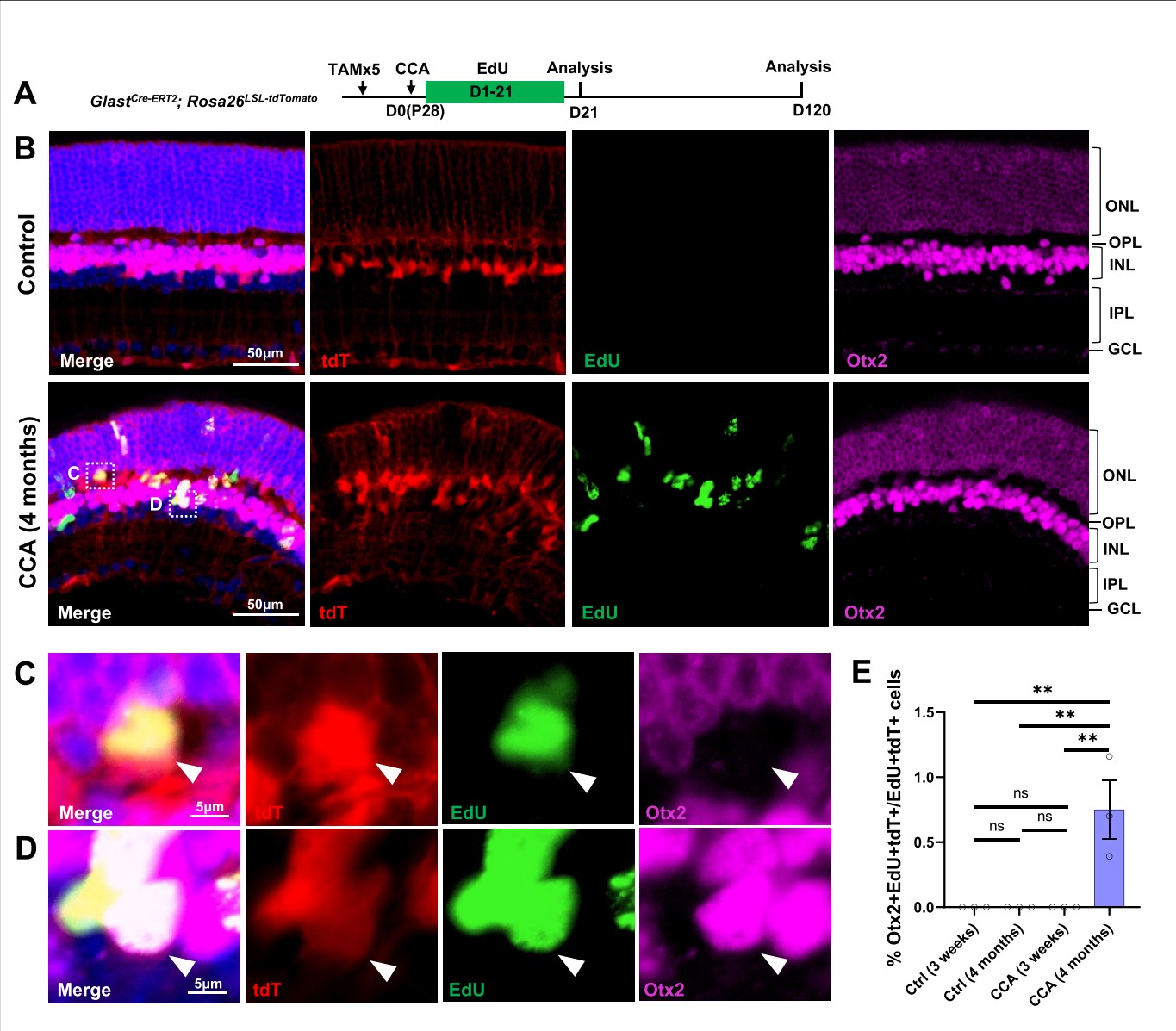

**Figure 6.** Cell cycle activator (CCA) induces de novo genesis of Otx2+ cells from Müller glia (MG). (**A**) Experimental design. (**B**) Representative retinal sections of *Glast^{Cre-ERT2}; Rosa26^{LSL-tdTomato}* mice without adeno-associated virus (AAV) injection (control) or mice at 4 months post-CCA injection. Sections were co-stained for EdU and Otx2. ONL, outer nuclear layer; OPL, outer plexiform layer; INL, inner nuclear layer; IPL, inner plexiform layer; GCL, ganglion cell layer. (**C–D**) Magnified views of the highlighted cells in (**B**). Arrowhead highlights a tdT+EdU+ MG in the OPL that is negative for Otx2 (**C**), while arrowhead highlights a tdT+EdU+Otx2+ MG in the INL. (**E**) Quantification of tdT+EdU+Otx2+ cells as a percentage of total tdT+EdU+ cells. n=3 mice, data are presented as mean ± SEM. ns = not significant, **p<0.01 (one-way ANOVA with Tukey's post hoc test).

The online version of this article includes the following figure supplement(s) for figure 6:

**Figure supplement 1.** Detection of Müller glia (MG)-derived HuC/D+ cells in the inner nuclear layer (INL) after cell cycle activator (CCA) treatment.

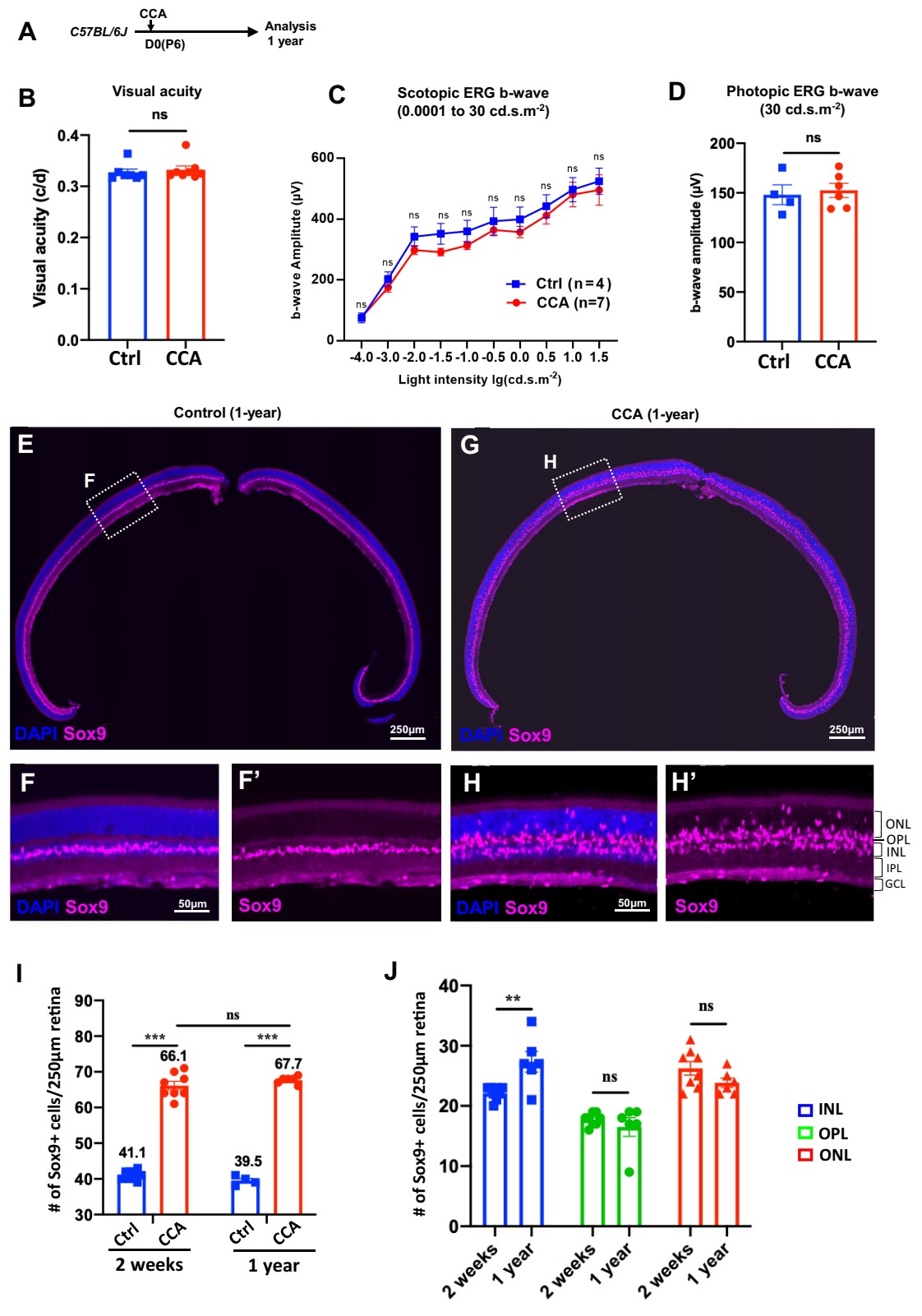

**Figure 7.** Cell cycle activator (CCA) does not lead to retinal neoplastic transformation. (**A**) Experimental design. (**B–C**) Optomotor and electroretinography (ERG) tests were performed on wild-type mice with one eye injected with CCA and the other eye as control (without AAV injection) at 1-year post-CCA injection. Visual acuity by the optomotor test (**B**), b-wave amplitudes of the scotopic ERG under different light intensity in (**C**), and b-wave amplitudes of the photopic ERG under 30 cd×s/m² in (**D**). Data are presented as mean ± SEM. ns = not significant (paired two-tailed Student's

*Figure 7 continued on next page*

*Figure 7 continued*

t-test) (**B–D**). (**E–H**) Immunostaining for MG marker Sox9. (**F, H**) Zoomed-in images of the boxed areas in (**E**) and (**G**). (**I**) Quantification of the numbers of Sox9+ cells in control retinas versus CCA-treated retinas at 2 weeks or 1-year post-CCA injection and in age-matched wild-type control retinas. Data are presented as mean ± SEM. ns = not significant, \*\*\*p<0.001 (one-way ANOVA with Tukey's post hoc test). (**J**) Quantification of the number of Sox9+ cells in each retinal layer. Data are presented as mean ± SEM. ns = not significant, \*\*p<0.01 (unpaired two-tailed Student's t-test).

The online version of this article includes the following figure supplement(s) for figure 7:

**Figure supplement 1.** Absence of TUNEL-positive Müller glia (MG) in the retina following cell cycle activator (CCA) treatment.

with TUNEL assay and did not observe any obvious TUNEL+ MG at 3 or 6 weeks post-CCA injection (*Figure 7—figure supplement 1*). Since we could only examine cell death at single time points, we cannot completely exclude the possibility that there were small numbers of MG-derived cells dying without being detected. The fact that the numbers of MG in the ONL and OPL did not decrease in the retinas harvested at 1-year post-CCA treatment suggests that these MG-derived cells persist without dying or losing the MG identity (*Figure 7J*).

## Discussion

MG proliferation is a key step toward MG-mediated regeneration in the retina. Previously, it was shown that activation of the canonical Wnt pathway by AAV-mediated overexpression of β-catenin stimulates MG proliferation through the Lin28/let7 pathway (*Yao et al., 2016*), and cyclin D1 is a direct target gene of both Wnt signaling and Lin28/let7 (*Li et al., 2012*; *Shtutman et al., 1999*; *Tetsu and McCormick, 1999*). Suppression of the Hippo pathway by transgenic expression of YAP[5SA], a dominant-active form of YAP, induces MG proliferation, accompanied by an increase in cyclin D1 expression (*Hamon et al., 2019*; *Rueda et al., 2019*). However, the Hippo and Wnt pathways have downstream genes besides cyclin D1 and serve different cellular functions. It remains unclear whether cyclin D1 activation is necessary or sufficient to drive MG proliferation in these contexts. In this study, we demonstrate that cyclin D1 overexpression alone leads to limited MG proliferation and that the combination of cyclin D1 overexpression and p27[Kip1] knockdown is the most potent strategy to drive MG proliferation in mouse retina without injury stimulus.

Our results showed that CCA effectively promoted MG proliferation in both P6 and adult mouse retinas; however, the timing of MG proliferation differed between the two age groups. Following CCA treatment on P6, MG proliferation started as early as day 3–4 post-CCA injection and largely finished by 2 weeks, peaking around 1 week after treatment (*Figure 2A*). In contrast, following CCA treatment on P28, MG proliferation started later, as not many mitotic MG had migrated to the ONL at 1-week post-CCA treatment (*Figure 2—figure supplements 2 and 4*). Moreover, proliferating MG were still present 3 weeks post-treatment, as shown by scRNA-seq analysis (*Figure 3B–E*). We speculate that the differences in MG proliferation time are due to variations in AAV transduction efficiencies of retinal cells at different ages. AAV[7m8] vectors delivered by intravitreal injections transduce MG more efficiently at P6 before the inner limiting membrane is fully developed. The level of cyclin D1 and p27[Kip1] may reach the thresholds permissive for MG cell cycle re-entry faster following CCA treatment on P6 compared to the same treatment in adult mice. This hypothesis is supported by the immunostaining results of both cyclin D1 and p27[Kip1] (*Figure 2G–I*, *Figure 2—figure supplement 2*, *Figure 2—figure supplement 3*, and *Figure 2—figure supplement 4*). This finding also emphasizes the importance of using high CCA titer (>5 × 10$^{13}$ genome copies/ml) in adult mice to effectively stimulate MG proliferation.

P27[Kip1] and cyclin D1 serve as critical regulators of cell cycle progression, and any abnormalities in their expression may impact cell division, potentially leading to tumor development. Previous studies have shown that mice lacking p27[Kip1] are approximately 30% larger and develop pituitary tumors spontaneously (*Fero et al., 1996*). Cyclin D1 is frequently upregulated in a significant fraction of human cancers, such as breast and respiratory tract tumors (*Callender et al., 1994*; *Zukerberg et al., 1995*). The strategy to stimulate MG proliferation for the purpose of retinal regeneration must be carefully weighed against the potential risk of tumorigenesis. In this study, we carefully evaluated the tumorigenic potential of CCA and made the following observations: First, the rate of MG proliferation gradually decreased and ceased over time (*Figure 2A*). Second, MG appeared to undergo only a single round of cell division (*Figure 2B–F*); and lastly, the retinal structure remained normal without signs

of neoplasia even 1 year post-treatment (*Figure 7*). The absence of continuous MG proliferation may be partly attributed to the non-integrating nature of the recombinant AAV genomes, which become diluted after cell division, leading to reduced transgene expression. Notably, the loss of cyclin D1 overexpression and the recovery of high-level p27$^{Kip1}$ did not occur simultaneously. The former was observed much earlier than the latter (*Figure 2*, *Figure 2—figure supplement 2*, *Figure 2—figure supplement 3*, and *Figure 2—figure supplement 4*). This is also supported by the scRNA-seq data, which showed that *Ccnd1* upregulation was lost in the MG 3 weeks after CCA treatment while *Cdkn1b* suppression was still evident (*Figure 3—figure supplement 6*). These findings suggest that the rapid cessation of cyclin D1 overexpression may be regulated by additional mechanisms. For instance, the antiproliferative gene *Btg2*, known to suppress cyclin D1 expression (*Wei et al., 2012*), was upregulated in the proliferating and reactivated MG (*Figure 3—figure supplement 6*). In summary, the risk of tumor formation associated with CCA is minimal.

Interestingly, we observed significant downregulation of IFN pathway genes in reactivated MG, a finding consistent with their known antiproliferative role in immune surveillance and tumor suppression (*Durbin et al., 1996*; *Kaplan et al., 1998*). The suppression of IFN pathway may reflect a mechanism by which CCA overcomes intrinsic barriers to proliferation. Our results align with *Rueda et al., 2019*, who reported that forced YAP$^{5SA}$ expression in NMDA-treated MG similarly downregulated IFN pathway genes in the proliferative MG (*Rueda et al., 2019*). Given that cyclin D1, a key Hippo pathway target, is upregulated in both paradigms, we propose a shared mechanism of IFN suppression. In addition to regulating cell cycle, cyclin D1 may indirectly control the transcription of *Stat1* and *Stat2* and/or other IFN pathway genes by interacting with other transcriptional cofactors (*McMahon et al., 1999*; *Zwijsen et al., 1998*). Notably, *Stat3* levels remained stable in the reactivated MG. STAT3 activation, which is primarily induced by cytokines (e.g. IL-6) and growth factors (e.g. EGF), promotes cell proliferation (*Hirano et al., 2000*), but it inhibits neurogenesis of mouse and avian MG (*Jorstad et al., 2020*; *Todd et al., 2016*; *Todd et al., 2020*). Sustained *Stat3* level in our system could explain both the proliferative competence of reactivated MG and their failure to differentiate to neurons. The divergent regulation of *Stat1/2* (suppressed) versus *Stat3* (sustained) highlights that these pathways are differentially modulated by the cell cycle regulators, warranting further investigation into their different functions in MG reprogramming.

The rods and rod-MG were two unexpected clusters in the scRNA-seq analysis. Upon close examination of the retina of *Glast$^{Cre-ERT2}$*; *Rosa26$^{LSL-tdTomato}$* mice without Tamoxifen induction, we found that very few rods (14 tdTomato+ rods in 128 whole retinal sections of eight mouse retina samples screened) were mislabeled by leaky tdTomato expression in a Cre-independent manner (*Figure 2—figure supplement 1*). The small rod clusters are likely the native rods mislabeled by tdTomato, appearing in all three groups at similar proportions. The rod-MG cluster, which was characterized by high levels of both MG and rod genes, was located between the rod cluster and the reactivated MG cluster (*Figure 3B–E*). This rod-MG cluster was enriched in the CCA and CCANT groups (*Figure 3D and E*). Initially, we speculated that this rod-MG cluster might represent MG that had upregulated rod genes. To investigate this, we performed RNA in situ hybridization to assess the levels of *Gnat1* and *Rho* mRNAs in MG freshly isolated from the mouse retina 3 weeks post-CCA treatment (*Figure 3—figure supplement 4*). Despite slight increases of *Gnat1* and *Rho* mRNA signals in the CCA-treated sample compared to the untreated control retina, some MG from the control retina also exhibited some signals of both genes, suggesting that these signals resulted from rod contamination (*Figure 3—figure supplement 4*). The enrichment of rod gene expression in CCA-treated MG, as indicated by the scRNA-seq data and RNA in situ hybridization data, may result from a greater number of MG migrating to the ONL and closely contacting with surrounding rods, leading to higher levels of rod contamination. Moreover, we did not observe many cells overexpressing neurogenic genes, such as *Ascl1, Neurog2*, and *Dll3*, or rod precursor gene *prdm1* (*Figure 3—figure supplement 6*). Therefore, we cautiously conclude that CCA alone does not reprogram MG toward a rod cell fate. The lack of MG reprogramming toward rod may be due to the sustained Notch signaling or Stat3 signaling in the reactivated MG (*Figure 3—figure supplement 6*).

Following cell division, the majority of the MG retained their glial identity, with around 1% of MG daughter cells expressing markers of bipolar or amacrine cells. This aligns with prior reports showing that a small subset of MG daughter cells expressed the markers of retinal interneurons after cell cycle reactivation either by inhibition of the Hippo pathway or by activation of Wnt signaling (*Rueda et al.,*

*2019*; *Yao et al., 2016*). The rarity of neurogenesis raises critical questions about intrinsic heterogeneity within the MG population. We speculate that a small subset of MG may exist in a 'primed' state, either expressing elevated levels of neurogenic factors or having a more permissive chromatin state for neurogenic gene expression, which predispose them to differentiate to neurons. However, the inefficiency of neurogenesis driven solely by cell cycle reactivation underscores a key translational barrier. To achieve functional regeneration, future strategies should combine MG cell cycle activation with neurogenic reprogramming factors (e.g. ASCL1) and/or suppression of anti-neurogenic pathways (e.g. Notch, STAT3). Such combinatorial approaches could redirect MG daughter cells from a default glial fate toward neuronal differentiation, offering a viable path for treating retinal degenerative diseases.

# Materials and methods

## Animals

Wild type C57BL/6J mice (Jax strain #000664), *Rosa26*[LSL-tdTomato] reporter mice (Jax strain #007914), *Rosa26*[LSL-Sun1:GFP] reporter mice (Jax strain #030952), and *Glast*[Cre-ERT2] reporter mice (Jax strain #012586) were purchased from the Jackson Laboratory and were kept on a 12 hr light/12 hr dark cycle. Both male and female mice were used in this study. Mice were randomly divided into control and treatment groups. All animal procedures performed were approved by the Hong Kong Department of Health under Animals Ordinance Chapter 340 ((24–234) in DH/HT&A/8/2/5 Pt.16). The protocol was approved by the Committee on the Ethics of Animal Experiments of City University of Hong Kong (AN-STA-00000283).

## AAV plasmids and vectors

The pAAV-GFAP-GFP vector plasmid was cloned by replacing the CMV promoter with a 681 bp ABC$_1$D region of the human *GFAP* gene promoter (*Lee et al., 2008*) into the pAAV-CMV-GFP vector (*Figure 1—figure supplement 1A*). cDNAs encoding mouse cyclin D1 were cloned into AAV plasmids by Gibson ligation. The AAV-shRNA vectors were cloned by replacing the GFP sequence with mCherry-shRNA in the pAAV-GFAP-GFP vector. Three shRNA sequences used in the study were GACTACACAAATCAGCGATTT (non-target shRNA), GCAAGTGGAATTTCGACTTTC (*p27* shRNA1), and GCTTGCCCGAGTTCTACTACA (*p27* shRNA2). pAAV-GFAP-cyclin D1-p27 shRNA1 and pAAV-GFAP-GFP-NT shRNA were cloned by replacing mCherry sequence with cyclin D1 or GFP from the pAAV-GFAP-mCherry-p27 shRNA1 or pAAV-GFAP-mCherry-NT shRNA, respectively (*Figure 1—figure supplement 2A*). The plasmids that were generated in this study will become available from Addgene. The pAAV rep/Cap 2/2 and adenovirus helper plasmids were obtained from the University of Pennsylvania Vector Core, Philadelphia. The pAAV7m8 plasmid (#64839) was purchased from Addgene.

AAV was produced in HEK293T cells (HCL4517; Thermo Scientific) by AAV vector plasmid, rep/cap packaging plasmid, and adenoviral helper plasmid co-transfection followed by iodixanol gradient ultracentrifugation. Purified AAVs were concentrated with Amicon 100K columns (EMD Millipore) to a final titer higher than 5×10$^{13}$ genome copies/ml. Protein gels were run to determine virus titers.

## Tamoxifen injection

Intraperitoneal injections of tamoxifen (Sigma) in corn oil were administered to induce the expression of Cre recombinase. Tamoxifen was given at a dosage of 50 mg/kg daily from P23 to P27 to induce reporter expression in the majority of MG and a single dosage of 15 mg/kg on P20 for sparsely labeling of MG.

## AAV injection

Intravitreal injection was performed using a pulled angled glass pipette controlled by a FemtoJet (Eppendorf). The tip of the needle was passed through the sclera at the equator, near the dorsal limbus of the eyeball, and entered the vitreous cavity. The injection volume of AAV (5×10$^{13}$ genome copies/ml) was 0.5 µl per eye for P6 injection and 1 µl per eye for adult mouse injection.

## In situ RNA hybridization on retinal sections

In situ RNA hybridization was performed on retinal sections using the RNAscope Multiplex Fluorescent Detection Reagents V2 kit (Advanced Cell Diagnostics) following the commercial protocols. In brief, retinas were dissected, 4% paraformaldehyde (PFA) fixed, dehydrated in sucrose solution, and embedded in optimal cutting temperature (OCT) medium. Retinas were cryosectioned into 20-µm-thick sections and mounted on SuperFrost Plus glass slides (Epredia). The uninjected eye and CCA-injected eye from the same animal were sectioned on the same slide and processed together. After OCT removal with PBS and further dehydration by ethanol, retinal sections were stained with GFP antibody (AB_2307313; Aves Labs) at 4°C overnight. After three times wash with PBST (PBS with 0.1% Tween-20), retinal sections were hybridized with RNA probes (Advanced Cell Diagnostics Cat. No. 426231-C2 for Mm *Glul*) at 40°C for 2 hr. Following in situ RNA hybridization steps, slides were stained using secondary antibodies (Jackson ImmunoResearch) and DAPI for 2 hr at room temperature. The fluorescent signals were visualized and captured using Nikon A1HD25 High speed and Large Field of View Confocal Microscope. The mRNA levels were quantified by measuring signal intensity level by ImageJ. Pixel intensity of MG treated with CCA was normalized to the average pixel intensity of MG in the uninjected eye of the same animal.

## RNA in situ hybridization on dissociated retinal cells

Fresh retina was dissociated in Papain (Worthington) and gently pipetting using a 1 ml pipette tip, and the dissociated retina cells were then filtered with 40 µm strainer (pluriStrainer). Filtered retinal cells were then seeded and cultured in a chamber slide (Thermo Fisher) at 37°C in a cell incubator. Following incubation, the cultured retinal cells were washed with PBS and fixed with 4% PFA. Dissociated cells were stained with GFP antibody (AB_2307313; Aves Labs) at 4°C overnight and then hybridized with RNA probes (Advanced Cell Diagnostics Cat.No. 474801-C3 for Mm *Rho*, 524881-C2 for Mm *Gnat1*) for 2 hr at 40°C for 2 hr. Following in situ RNA hybridization steps, slides were stained using secondary antibodies (Jackson ImmunoResearch) and DAPI for 2 hr at room temperature. The fluorescent signals were visualized and captured using Nikon A1HD25 High speed and Large Field of View Confocal Microscope. The mRNA levels were quantified by counting the numbers of RNA dots using ImageJ by another experimenter with group masked.

## Quantitative PCR

RNA was extracted from whole retina using TRIzol (Thermo Fisher Scientific) followed by the Quick-RNA MicroPrep Kit (Zymo Research). RNAs were converted to cDNA using a PrimeScript RT reagent kit with gDNA Eraser (Takara Bio). qPCR was performed using the PowerUp Sybr Green Master Mix (Thermo Fisher Scientific) on QuantStudio 3 Real-Time PCR stems (Applied Biosystems). Gapdh was used as the normalizing control. qPCR primers are listed below:

```
p27_qPCR_F: TCAAACGTGAGAGTGTCTAACG
p27_qPCR_R: CCGGGCCGAAGAGATTTCTG
CycD1_qPCR_F: CCCAACAACTTCCTCTCCTG
CycD1_qPCR_R: TCCAGAAGGGCTTCAATCTG
Gapdh_qPCR_F: AGGTCGGTGTGAACGGATTTG
Gapdh_qPCR_R: TGTAGACCATGTAGTTGAGGTCA
```

## Immunohistochemistry

Retinas were dissected and fixed in 4% formaldehyde in PBS for 30 min at room temperature and sectioned at 20 µm thickness by cryostat. Retinal sections were blocked in 5% BSA in PBST (PBS with 0.1% Triton X-100), stained with primary antibodies at 4°C overnight, and washed three times with PBST. Primary antibodies used in this study included rabbit anti-p27[kip1](1:200, PA5-16717; Thermo Fisher); rabbit anti-cyclin D1 antibody (1:300, 26939-1-AP; Proteintech), goat anti-Otx2 antibody (1:200, AF1979; R&D Systems), mouse anti-HuC/D antibody (1:200, A21271; Thermo Fisher), rabbit anti-GFAP antibody (1:500, Z0334, DAKO), goat anti-Sox2 antibody (1:500, AF2018, R&D Systems), and rabbit anti-Sox9 antibody (1:1000, AB5535; Millipore). Sections were stained using secondary antibodies (Jackson ImmunoResearch) for 2 hr at room temperature. Cell nuclei were counterstained with

DAPI (Sigma). TUNEL was performed using an TUNEL BrightRed Apoptosis Detection Kit (Vazyme) according to the manufacturer's protocol.

## EdU incorporation and BrdU detection assay

EdU (50 mg/kg, Abcam ab146186) or BrdU (100 mg/kg, Abcam 142567) was injected intraperitoneally to label the cells in the S phase. EdU staining was performed using the Click-iT EdU Alexa Fluor Imaging Kit (C10337). For BrdU detection, the retinal sections were incubated with 2 M HCl for 1 hr at room temperature. The sections were rinsed with PBST and incubated with a blocking buffer containing 5% BSA in PBST for 2 hr at room temperature. Primary antibody mouse anti-BrdU (1:300, ab8152; Abcam) was stained overnight at 4°C, and secondary antibodies (Jackson ImmunoResearch) were stained for 2 hr at room temperature.

## Single-cell RNA-seq

### Library preparation and sequencing of single cells

The FACS-sorted cells from each sample were filtered through a 40 μm strainer (pluriStrainer) and processed with Chromium Next GEM Single Cell 3′ GEM, Library & Gel Bead Kit v3.1 (1000269), Chromium Next GEM Chip G Single Cell Kit (1000127) and Chromium Controller (10x Genomics) according to the manufacturer's protocol. The constructed libraries were sent to Novogene (Beijing) for NovaSeq paired-end 150 bp sequencing and produced 330 Gb of raw data.

### Preprocessing, filtering, and clustering of scRNA data

Sequencing results were preprocessed with 10x Genomics Cell Ranger 6.1.1 (*Zheng et al., 2017*) for demultiplexing, barcode assignment, and unique molecular identifier (UMI) quantification using a standard pipeline for 10× for each of the three samples. We used indexed mouse genome mm10 with added tdTomato and GFP transgenes as a reference. Reads mapping to introns covered more than 34% of all reads. Therefore, they were included in feature counts using 'include_introns' option in Cell Ranger count step of the pipeline. Cell Ranger aggregation was applied for merging runs from three samples.

Quality control, filtering, dimensional reduction, and clustering of the data were carried out using the Seurat package in R (*Stuart et al., 2019*). The cells with less than 2000 expressed features, total UMI count higher than 100,000, and percentage of mitochondrial transcripts more than 15% were disregarded. Features of the Y chromosome, as well as *Xist* and *Tsix* genes of the X chromosome, were excluded to avoid confusing effects of gender.

Further filtering of cells was carried out by only including cells expressing tdTomato (*Figure 3— figure supplement 1*). Potential cell duplicates were estimated using DoubletFinder method (*McGinnis et al., 2019*) implemented in R (with parameter settings PC = 10, pK = 0.28, pN = 0.3). The threshold for the expected doublet formation rate was set to 10% to reflect the assumption that cell duplicates can appear relatively commonly in the MG cells. Detected duplicates were excluded from the data (*Figure 3—figure supplement 1G and H*).

For the remaining cells, Uniform Manifold Approximation and Projection (UMAP) dimension reduction based on eight principal components (PCs) was applied, and cells were clustered using the graphical clustering method in Seurat. Cell types were identified using known marker genes (*Clark et al., 2019*; *Hoang et al., 2020*). After removing doublets, cell types in clusters 4–9 other than MG, Rod-MG, and Rod included only a few cells, and 10–30% of the cells were from the control sample, suggesting that these cells are likely to be contaminated (*Figure 3—figure supplement 1I and J*). After excluding cells in clusters 4–9, the new UMAP and clustering revealed two subclusters of Rod-MG cells (clusters C3-C4, *Figure 3—figure supplement 2A*). The cells in the subcluster C4 exhibited similar proportional contributions from all three groups and expressed lower level of rod-specific genes such as *Rho* (*Figure 3—figure supplement 2B and C*). These cells were again defined as contaminated and excluded from the data. The final UMAP was constructed using seven PCs, and the same PCs were used for clustering of cells (*Figure 3B*).

### Analysis of scRNA-seq data

The cell cycle state of each cell was defined by the Cell Cycle Scoring function in the Seurat R package. The cells with G2/M and S scores lower than 0.1 were defined to be proliferating (*Figure 3—figure*

*supplement 2D and E*). Differential gene expression analysis between different cell types or three treatment groups was performed using DESeq2 (*Supplementary file 1*; *Love et al., 2014*). Enrichment analysis of top 200 up- and downregulated genes was performed for Gene Ontologies (GO), KEGG, and Reactome pathways using Gprofiler2 R interface to g:Profiler (*Raudvere et al., 2019*).

### The optomotor and ERG tests

Mouse visual acuity was measured using an Optometry System (Cerebral Mechanics Inc) following the published protocol (*Douglas et al., 2005*). Testing was done with a grating of 12 degrees/s drifting speed and 100% contrast. The injected right eyes (CCA-treated) and uninjected left eyes (Ctrl) were tested independently for counterclockwise and clockwise grating rotations, respectively. A staircase procedure was used, in which the observer tested low to high visual acuity. Each animal was tested for about 10–15 min per session.

ERG was measured using an Espion E3 System (Diagonsys LLC Inc) as previously described (*Hoang et al., 2023*). Animals were dark-adapted overnight before the test. After inducing anesthesia with a ketamine:xylazaine injection intraperitoneally, the pupils of both eyes were dilated using Mydrin-P Ophthalmic Solution. Once the pupils were fully dilated, the eyes were maintained moist using a topical gel. The measurement electrodes were then applied to the cornea of both eyes, while the ground electrodes were attached to the mouth and tail. All these steps were performed in the dark-room under dim red light. For scotopic ERG recordings, a multiple 530 nm light with different intensities (increments from 0.01 to 30 cd.s/m$^2$) were elicited to stimulate scotopic responses in a specific time interval. For photopic ERG recordings, 5 min exposure under 10 cd.s/m$^2$ light intensity was adopted to inhibit the rod function. The photopic response was measured by multiple flashes of 30 cd.s/m$^2$ intensity in the illuminated background (10 cd.s/m$^2$). The average amplitude and implicit time of a- and b-wave were recorded and exported for further analysis. The b-wave amplitude of the step-wise scotopic responses and that of photopic response at 30 cd.s/m$^2$ were shown in figure.

### Statistics

Data were presented as mean ± SEM in all figures. Sample sizes and statistical analysis were indicated for each experiment in figure legend. ANOVA with Tukey's test was performed to compare multiple groups and Student's t-test to compare two groups. A p-value<0.05 was considered statistically significant. GraphPad Prism was used to perform statistical analysis and make figures.

## Acknowledgements

This research was funded by Hong Kong Research Grants Council Project (11103819, 11102922, and 11100723), Hong Kong Health and Medical Research Fund Project (05160276 and 06172466), TUNG Biomedical Sciences Foundation, and Ming Wai Lau Center for Reparative Medicine Research Associate Program.

## Additional information

### Funding

| Funder | Grant reference number | Author |
| --- | --- | --- |
| Research Grants Council, University Grants Committee | 11103819 | Wenjun Xiong |
| Research Grants Council, University Grants Committee | 11102922 | Wenjun Xiong |
| Research Grants Council, University Grants Committee | 11100723 | Wenjun Xiong |
| Health and Medical Research Fund | 05160276 | Wenjun Xiong |

| Funder | Grant reference number | Author |
|---|---|---|
| Health and Medical Research Fund | 06172466 | Wenjun Xiong |
| TUNG Biomedical Sciences Foundation | | Wenjun Xiong |
| Ming Wai Lau Center for Reparative Medicine Research Associate Program | | Wenjun Xiong |

The funders had no role in study design, data collection and interpretation, or the decision to submit the work for publication.

## Author contributions

Zhifei Wu, Conceptualization, Data curation, Formal analysis, Methodology, Writing – original draft; Baoshan Liao, Data curation, Formal analysis, Validation, Methodology, Writing – original draft, Writing – review and editing; Julia Ying, Formal analysis; Jan Keung, Zongli Zheng, Methodology; Virpi Ahola, Software, Formal analysis, Methodology, Writing – original draft; Wenjun Xiong, Conceptualization, Resources, Supervision, Funding acquisition, Investigation, Methodology, Writing – original draft, Project administration, Writing – review and editing

## Author ORCIDs

Baoshan Liao http://orcid.org/0009-0007-6599-624X
Julia Ying http://orcid.org/0009-0006-4574-1058
Zongli Zheng http://orcid.org/0000-0003-4849-4903
Wenjun Xiong https://orcid.org/0000-0001-6836-2807

## Ethics

All animal procedures performed were approved by the Hong Kong Department of Health under Animals Ordinance Chapter 340 ((24-234) in DH/HT&A/8/2/5 Pt.16). The protocol was approved by the Committee on the Ethics of Animal Experiments of City University of Hong Kong (AN-STA-00000283).

Reviewer #1 (Public review): https://doi.org/10.7554/eLife.100904.3.sa1
Reviewer #2 (Public review): https://doi.org/10.7554/eLife.100904.3.sa2
Author response https://doi.org/10.7554/eLife.100904.3.sa3

# Additional files

## Supplementary files

Supplementary file 1. Differential gene expression analysis between different cell types or three treatment groups. Differential gene expression analysis between different cell types or three treatment groups was performed using DESeq2. All genes with counts>0 in the test were considered. No other thresholds were used.

MDAR checklist

## Data availability

ScRNA-Sequencing data have been deposited in GEO under accession codes GSE225142.

The following dataset was generated:

| Author(s) | Year | Dataset title | Dataset URL | Database and Identifier |
|---|---|---|---|---|
| Wu Z, Liao B, Ying J, Keung J, Zheng Z, Ahola V, Xiong W | 2024 | Simultaneous cyclin D1 overexpression and p27kip1 knockdown enable robust Müller glia cell cycle reactivation in uninjured mouse retina | https://www.ncbi.nlm.nih.gov/geo/query/acc.cgi?acc=GSE225142 | NCBI Gene Expression Omnibus, GSE225142 |

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
