## [Editor Report · eLife Assessment]

This manuscript presents a potentially **important** strategy for stimulating mammalian Müller glia to proliferate in vivo by manipulating cell cycle components. The results are **convincing** that a large number of Müller glia can be induced to re-enter the cell cycle without a damage stimulus. These findings are likely to appeal to retinal biologists and neuroscientists in general.

---

## [Referee Report · Reviewer #1 (Public review)]

Summary:

In this manuscript, Wu et al. introduce a novel approach to reactivate the Muller glia cell cycle in the mouse retina by simultaneously reducing p27Kip1 and increasing cyclin D1 using a single AAV vector. The approach effectively promotes Muller glia proliferation and reprogramming without disrupting retinal structure or function. Interestingly, reactivation of the Muller glia cell cycle downregulates IFN pathway, which may contribute to the induced retinal regeneration. The results presented in this manuscript may offer a promising approach for developing Müller glia cell-mediated regenerative therapies for retinal diseases.

Comments on revisions:

The authors have revised the manuscript and addressed my concerns.

---

## [Referee Report · Reviewer #2 (Public review)]

This manuscript by Wu, Liao et al. reports that simultaneous knockdown of P27Kip1 with overexpression of Cyclin D can stimulate Muller glia to re-enter the cell cycle in the mouse retina. There is intense interest in reprogramming mammalian muller glia into a source for neurogenic progenitors, in the hopes that these cells could be a source for neuronal replacement in neurodegenerative diseases. Previous work in the field has shown ways in which mouse Muller glia can be neurogenically reprogrammed and these studies have shown cell cycle re-entry prior to neurogenesis. In other works, typically, the extent of glial proliferation is limited, and the authors of this study highlight the importance of stimulating large numbers of Muller glia to re-enter the cell cycle with the hopes they will differentiate into neurons.

The authors have satisfactorily responded to all my previous reviewer comments. The authors have significantly improved their imaging quality in Figure 1 and 4. The authors have admirably re-considered their FISH and scRNA-seq data and performed critical control experiments. They now provide a more nuanced interpretation of their data by removing reference to MG-inducing rod genes which is now interpreted as ambient contamination. Taken together, this manuscript now provides strong evidence of a viral way to induce large numbers of MG to re-enter the cell cycle without a damage stimulus.

---

## [Author Response]

The following is the authors’ response to the original reviews.

**Public Reviews:**

**Reviewer #1 (Public Review):**
Summary:In this manuscript, Wu et al. introduce a novel approach to reactivate the Muller glia cell cycle in the mouse retina by simultaneously reducing p27Kip1 and increasing cyclin D1 using a single AAV vector. The approach effectively promotes Muller glia proliferation and reprograming without disrupting retinal structure or function. Interestingly, reactivation of the Muller glia cell cycle downregulates IFN pathway, which may contribute to the induced retinal regeneration. The results presented in this manuscript may offer a promising approach for developing Müller glia cell-mediated regenerative therapies for retinal diseases.Strengths:The data are convincing and supported by appropriate, validated methodology. These results are both technically and scientifically exciting and are likely to appeal to retinal specialists and neuroscientists in general.Weaknesses:There are some data gaps that need to be addressed.(1) Please label the time points of AAV injection, EdU labeling, and harvest in Figure 1B.

We thank the reviewer for highlighting the lack of clarity in our experimental design. We have labeled all experiment timelines in the figures where appropriate in the revised version.

(2) What fraction of Müller cells were transduced by AAV under the experimental conditions?

We apologize for not clearly explaining the AAV transduction effeciency. AAV transduction efficiency was not uniform across the retinas. The retinal region adjacent to the optic nerve exhibits a transduction efficiency of nearly 100%. In contrast, the peripheral retina shows a lower transduction efficiency compared to the central region. The representative retinal sections with typical infection pattern are shown in Supplementary figure 4. The quantification of Edu+ MG or other markers was conducted in a 250 µm region with the highest efficiency. For scRNA-seq experiment, retinal regions with high AAV transduction efficiency were dissected with the aid of a control GFP virus.

(3) It seems unusually rapid for MG proliferation to begin as early as the third day after CCA injection. Can the authors provide evidence for cyclin D1 overexpression and p27 Kip1 knockdown three days after CCA injection?

We included the data that GFP expression is evident at 3 days post AAV-GFP-GFP injection (Supplementary Fig. 1B). Additionally, we performed immunostaining and confirmed cyclin D1 overexpression at 3 days post CCA injection (Fig. 2E) as well as qPCR analysis to confirm cyclin D1 overexpression and p27kip1 knockdown at the same time point (Supplementary Fig. 5).

(4) The authors reported that MG proliferation largely ceased two weeks after CCA treatment. While this is an interesting finding, the explanation that it might be due to the dilution of AAV episomal genome copies in the dividing cells seems far-fetched.

We agree with the reviewer that dilution of AAV episomal genomes is unlikely to be the sole reason for the stop of MG proliferation. By staining cyclin D1 at various days post CCA injection, we found that cyclin D1 is immediately downregulated in the mitotic MG undergoing interkinetic nuclear migration to the outer nuclear layer (Fig. 2G-I). In contrast, the effect of p27^kip1^ knockdown by CCA lasted longer (Supplementary Figure 9-10). It is possible that other anti-proliferative genes are involved in the immediate downregulation of Cyclin D1.

**Reviewer #2 (Public Review):**
This manuscript by Wu, Liao et al. reports that simultaneous knockdown of P27Kip1 with overexpression of Cyclin D can stimulate Muller glia to re-enter the cell cycle in the mouse retina. There is intense interest in reprogramming mammalian muller glia into a source for neurogenic progenitors, in the hopes that these cells could be a source for neuronal replacement in neurodegenerative diseases. Previous work in the field has shown ways in which mouse Muller glia can be neurogenically reprogrammed and these studies have shown cell cycle re-entry prior to neurogenesis. In other works, typically, the extent of glial proliferation is limited, and the authors of this study highlight the importance of stimulating large numbers of Muller glia to re-enter the cell cycle with the hopes they will differentiate into neurons. While the evidence for stimulating proliferation in this study is convincing, the evidence for neurogenesis in this study is not convincing or robust, suggesting that stimulating cell cycle-reentry may not be associated with increasing regeneration without another proneural stimulus.Below are concerns and suggestions.Intro:(1) The authors cite past studies showing "direct conversion" of MG into neurons. However, these studies (PMID: 34686336; 36417510) show EdU+ MG-derived neurons suggesting cell cycle re-entry does occur in these strategies of proneural TF overexpression.

We thank the reviewer for pointing this out. We have revised the statement to "MG reprogramming".

(2) Multiple citations are incorrectly listed, using the authors first name only (i.e. Yumi, et al; Levi, et al;). Studies are also incompletely referenced in the references.

We apologize for the mistakes in reference. We have corrected the reference mistakes in the revised version.

Figure 1:(3) When are these experiments ending? On Figure 1B it says "analysis" on the end of the paradigm without an actual day associated with this. This is the case for many later figures too. The authors should update the paradigms to accurately reflect experimental end points.

We thank the reviewer for highlighting the lack of clarity in our experimental design. We have labeled all experiment timelines in the figures where appropriate in the revised version.

(4) Are there better representative pictures between P27kd and CyclinD OE, the EdU+ counts say there is a 3 fold increase between Figure 1D&E, however the pictures do not reflect this. In fact, most of the Edu+ cells in Figure 1E don't seem to be Sox9+ MG but rather horizontally oriented nuclei in the OPL that are likely microglia.

Thanks to the reviewer for pointing this out. We have replaced the image of cyclin D1 OE retina which a more representative image.

(5) Is the infection efficacy of these viruses different between different combinations (i.e. CyclinD OE vs. P27kd vs. control vs. CCA combo)? As the counts are shown in Figure 1G only Sox9+/Edu+ cells are shown not divided by virus efficacy. If these are absolute counts blind to where the virus is and how many cells the virus hits, if the virus efficacy varies in efficiency this could drive absolute differences that aren't actually biological.

Rule out the possibility that the differences in MG proliferation across groups are due to variations in viral efficacy, we have examined the p27^kip1^ knockdown and cyclin D1 overexpression efficiencies for all four groups by qPCR analysis. The result showed that cyclin D1 overexpression efficiency by AAV-GFAP-Cyclin D1 virus alone or P27 knockdown efficiency by AAV-GFAP-mCherry-p27kip1 shRNA1 is comparable to, if not even higher than, those by CCA virus (Supplementary Fig 5). Therefore, the virus efficacy cannot explain the drastic increase in MG proliferation by CCA.

As the central retina usually had 100% infection efficacy (Supplementary Fig. 4), we quantified the Edu+Sox9+ cell number in the 250µm regions next to the optic nerve.

(6) According to the Jax laboratories, mice aren't considered aged until they are over 18months old. While it is interesting that CCA treatment does not seem to lose efficacy over maturation I would rephrase the findings as the experiment does not test this virus in aged retinas.

Thank you to the reviewer for bringing this to our attention. We have changed to “older adult mice” in our revised manuscript.

(7) Supplemental Figure 2c-d. These viruses do not hit 100% of MG, however 100% of the P27Kip staining is gone in the P27sh1 treatment, even the P27+ cell in the GCL that is likely an astrocyte has no staining in the shRNA 1 picture. Why is this?

We have replaced the images in Supplementary Fig. 2B-D.

Figure 2(8) Would you expect cells to go through two rounds of cell cycle in such a short time? The treatment of giving Edu then BrdU 24 hours later would have to catch a cell going through two rounds of division in a very short amount of time. Again the end point should be added graphically to this figure.

We thank the reviewer for the comment. We repeated the Edu/BrdU colabelling experiment with extended periods of Edu/BrdU injections. Based on the result of the MG proliferation time course study (Fig. 2A), we injected 5 times of Edu from D1 to D5 and 5 times of BrdU from D6 to D10 post-CCA injection, which covered the major phase of MG proliferation (Fig. 2B-C). Consistent with the previous findings, we did not observe any BrdU&EdU double positive MG cells.

Additionally, we showed that cyclin D1 overexpression immediately ceased in migrating mitotic MG (Fig. 2G-I), which may explain why CCA-treated MG do not progress to the second round of cell division.

Figure 3(9) I am confused by the mixing of ratios of viruses to indicate infection success. I know mixtures of viruses containing CCA or control GFP or a control LacZ was injected. Was the idea to probe for GFP or LacZ in the single cell data to see which cells were infected but not treated? This is not shown anywhere?

The virus infection was not uniform across the entire retina (Supplementary Fig. 4). To mark the infection hotspots, we added 10% GFP virus to the mixture. Regions of the retina with low infection efficiency were removed by dissection and excluded from the scRNA-seq analysis. Therefore, we assumed that the vast majority of MG were infected by CCA. We apologize for not clearly explaining this methodological detail in the original text. We have added the experimental design to Fig. 3A and revised the result part (line 191-196) accordingly.

(10) The majority of glia sorted from TdTomato are probably not infected with virus. Can you subset cells that were infected only for analysis? Otherwise it makes it very hard to make population judgements like Figure 3E-H if a large portion are basically WT glia.

This question is related to the last one. Since the regions with high virus infection efficiency were selectively dissected and isolated for analysis, the CCA-infected MG should constitute the vast majority of MG in the scRNA-seq data.

(11) Figure 3C you can see Rho is expressed everywhere which is common in studies like this because the ambient RNA is so high. This makes it very hard to talk about "Rod-like" MG as this is probably an artifact from the technique. Most all scRNA-seq studies from MG-reprogramming have shown clusters of "rods" with MG hybrid gene expression and these had in the past just been considered an artifact.

We agree with the reviewer that the high rod gene expression in the rod-MG cluster is an artifact. We have performed multiple rounds of RNA in situ hybridization on isolated MG nuclei. The counts of Gnat1 and Rho mRNA signal are largely overlapped between the two samples with and without CCA treatment (Supplementary Fig 14). Some MG in the control retinas without CCA treatment had up to 7 or 8 dots per cell, suggesting contamination of attached rod cell debris during retina dissociation (Supplementary Fig 14). Therefore, the result did not support that rod-MG is a reprogrammed MG population with rod gene upregulation.

(12) It is mentioned the "glial" signature is downregulated in response to CCA treatment. Where is this shown convincingly? Figure H has a feature plot of Glul, which is not clear it is changed between treatments. Otherwise MG genes are shown as a function of cluster not treatment.

We have added box plots of several MG-specific genes to illustrate the downregulation of the glial signature in the relevant cell cluster in the revised manuscript (Supplementary Fig. 15).

Figure 4(13) The authors should be commended for being very careful in their interpretations. They employ the proper controls (Er-Cre lineage tracing/EdU-pulse chasing/scRNA-seq omics) and were very careful to attempt to see MG-derived rods. This makes the conclusion from the FISH perplexing. The few puncta dots of Rho and GNAT in MG are not convincing to this reviewer, Rho and GNAT dots are dense everywhere throughout the ONL and if you drew any random circle in the ONL it would be full of dots. The rigor of these counts also comes into question because some dots are picked up in MG in the INL even in the control case. This is confusing because baseline healthy MG do not express RNA-transcripts of these Rod genes so what is this picking up? Taken together, the conclusion that there are Rod-like MG are based off scRNA-seq data (which is likely ambient contamination) and these FISH images. I don't think this data warrants the conclusion that MG upregulate Rod genes in response to CCA.

Given the results of RNA in situ hybridization on isolated MG, we revisited the result of the RNA in situ hybridization on retinal sections as well. We performed RNA in situ in the retinal section at 1 week post CCA treatment, expecting to see lower Gnat1 and Rho signals in the ONL-localizing MG compared to 3 weeks and 4 months post CCA treatment. However, we observed similar levels across all three time points (data not shown). The lack of dynamic changes in rod gene expression levels also suggests contamination from tightly surrounding neighboring rods. Consequently, we have reinterpreted the scRNA-seq and RNA FISH data and withdrawn the conclusion that MG upregulated rod genes after CCA treatment. We thank the reviewer for pointing out this potential issue and helping us avoid an incorrect conclusion.

Figure 5(14) Similar point to above but this Glul probe seems odd, why is it throughout the ONL but completely dark through the IPL, this should also be in astrocytes can you see it in the GCL? These retinas look cropped at the INL where below is completely black. The whole retinal section should be shown. Antibodies exist to GS that work in mouse along with many other MG genes, IHC or western blots could be done to better serve this point.

We have replaced the images in Figure 4 in the revised manuscript. Additionally, we have performed the Sox9 antibody staining to demonstrate partial MG dedifferentiation following CCA treatment (Figure 5).

Figure 6(15) Figure 6D is not a co-labeled OTX2+/ TdTomato+ cell, Otx2 will fill out the whole nucleus as can be seen with examples from other MG-reprogramming papers in the field (Hoang, et al. 2020; Todd, et al. 2020; Palazzo, et al. 2022). You can clearly see in the example in Figure 6D the nucleus extending way beyond Otx2 expression as it is probably overlapping in space. Other examples should be shown, however, considering less than 1% of cells were putatively Otx2+, the safer interpretation is that these cells are not differentiating into neurons. At least 99.5% are not.

We have replaced the image of Otx2+ Tdt+ Edu+ cell, which shows the whole nucleus filled with strong Otx2 staining.

(16) Same as above Figure 6I is not convincingly co-labeled HuC/D is an RNA-binding protein and unfortunately is not always the clearest stain but this looks like background haze in the INL overlapping. Other amacrine markers could be tested, but again due to the very low numbers, I think no neurogenesis is occurring.

Since we didn’t find HuC/D+Tdt+EdU+ cells at 3 weeks post CCA treatment, we believe that the weak HuC/D+ staining in the MG daughter cells at 4 months is not background, but rather reflects an incomplete neurogenic switch. This suggests that the process of neurogenesis may be ongoing but not fully realized within the observed timeframe without additional stimuli.

(17) In the text the authors are accidently referring to Figure 6 as Figure 7.

We thank the reviewer for pointing out the mistake. We will correct the mistake in the revised manuscript.

Figure 7(18) I like this figure and the concept that you can have additional MG proliferating without destroying the retina or compromising vision. This is reminiscent of the chick MG reprogramming studies in which MG proliferate in large numbers and often do not differentiate into neurons yet still persist de-laminated for long time points.General:(19) The title should be changed, as I don't believe there is any convincing evidence of regeneration of neurons. Understanding the barriers to MG cell-cycle re-entry are important and I believe the authors did a good job in that respect, however it is an oversell to report regeneration of neurons from this data.

We thank the reviewer for the suggestion. We have changed the title to “Simultaneous cyclin D1 overexpression and p27kip1 knockdown enable robust Müller glia cell cycle reactivation in uninjured mouse retina” in the revised manuscript.

(20) This paper uses multiple mouse lines and it is often confusing when the text and figures switch between models. I think it would be helpful to readers if the mouse strain was added to graphical paradigms in each figure when a different mouse line is employed.

We have labeled the mouse lines used in each experiment in the figures where appropriate.